# GENERALIZATION OF RLVR USING CAUSAL REASONING AS A TESTBED

**Brian Lu** [1*]   **Hongyu Zhao** [2]   **Shuo Sun** [1]   **Hao Peng** [3]   **Rui Ding** [4]   **Hongyuan Mei** [5]
[1]Johns Hopkins University    [2]University of Maryland, College Park
[3]University of Illinois at Urbana-Champaign    [4]Microsoft Research Asia
[5]Toyota Technological Institute at Chicago

## ABSTRACT

Reinforcement learning with verifiable rewards (RLVR) has emerged as a promising paradigm for post-training large language models (LLMs) on complex reasoning tasks. Yet, the conditions under which RLVR yields robust generalization remain underexplored. This paper provides an empirical study of RLVR generalization in the setting of probabilistic inference over causal graphical models. This setting offers two natural axes along which to examine generalization: (i) the level of the probabilistic query—associational, interventional, or counterfactual—and (ii) the structural complexity of the query, measured by the size of its relevant subgraph. We construct a dataset of causal graphs and queries spanning these difficulty axes and fine-tune Qwen-2.5-Instruct models using RLVR or supervised fine-tuning (SFT). We vary both the model scale (3B-32B) and the query level included in training. We find that RLVR yields stronger within-level and across-level generalization than SFT, but only for specific combinations of model size and training query level. Further analysis shows that RLVR's effectiveness depends on the model's initial reasoning competence. With sufficient initial competence, RLVR improves an LLM's marginalization strategy and reduces errors in intermediate probability calculations, producing substantial accuracy gains, particularly on more complex queries. These results show that RLVR can improve specific causal reasoning subskills, with its benefits emerging only when the model has sufficient initial competence. Our code and data is available at `https://github.com/zhichul/rlcausal`.

## 1 INTRODUCTION

Reinforcement learning with verifiable rewards (RLVR) (Lambert et al., 2025; DeepSeek-AI et al., 2025) is a promising paradigm for post-training large language models (LLMs) on complex reasoning tasks. RLVR leverages automatic correctness signals from domains equipped with reliable verifiers, and has enabled substantial progress in mathematical problem solving (Shao et al., 2024; Lambert et al., 2025; DeepSeek-AI et al., 2025), formal theorem proving (Xin et al., 2024; Ren et al., 2025; Wang et al., 2025), code generation (Le et al., 2022; Liu & Zhang, 2025), and in biomedical and chemistry applications (Biomni & Sky RL, 2025; Narayanan et al., 2025). Despite rapid progress across diverse domains, the conditions under which LLMs trained with RLVR exhibit reliable generalization beyond their training data remain underexplored.

Recent work has begun to examine the generalization behavior of reinforcement-learning fine-tuning (RL) relative to supervised fine-tuning (SFT) or hybrid approaches (Chu et al., 2025; Chen et al., 2025; Swamy et al., 2025; Qiu et al., 2025). Particularly relevant is Chu et al. (2025), which evaluates the generalization of RLVR and SFT on novel variants of text and visual reasoning tasks. Our work differs from these prior work by focusing on a challenging and essential task: causal inference.

Causal inference provides a structured setting for examining RLVR generalization, because its three levels of inference—associational, interventional, and counterfactual, known collectively as the causal ladder (Bareinboim et al., 2022; Pearl & Mackenzie, 2018)—form a hierarchy that supports

---

*Correspondence to: zlu39@jhu.edu.

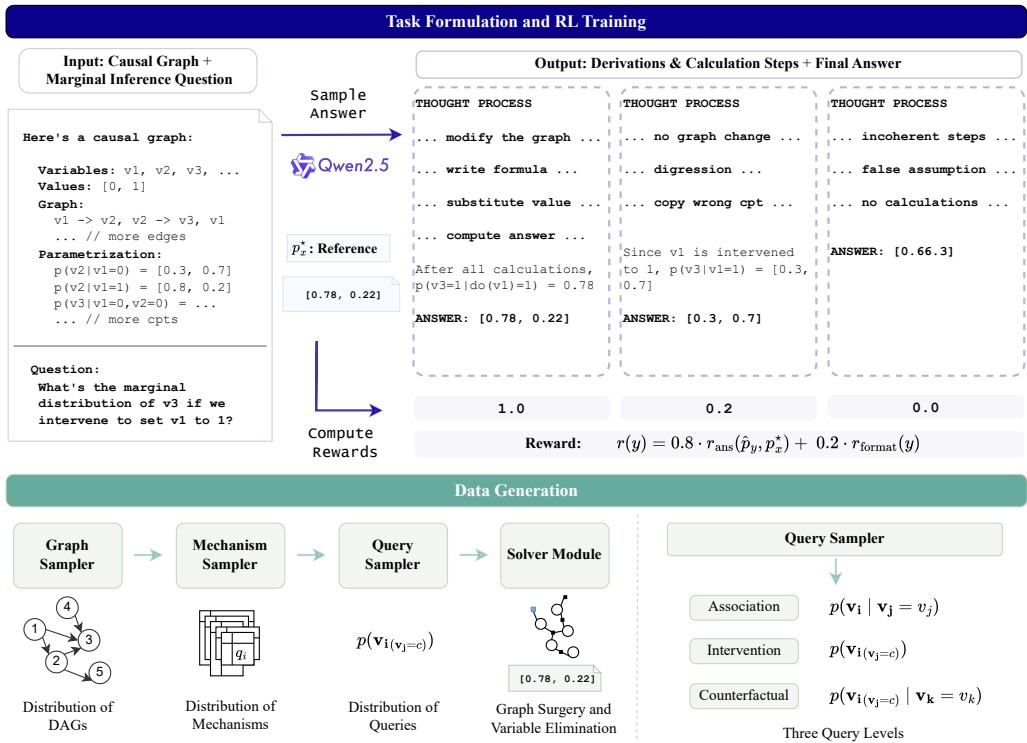

Figure 1: Top: Our causal inference task for investigating generalization of RLVR (see section 2), system prompt (fig. 8) omitted for space. Bottom Left: Generative process for sampling task instances, and solver for computing the reference (see section 3). Bottom Right: We generate association, intervention, and counterfactual queries to study RLVR's within-/across-level generalization.[3]

both within- and across-level generalization tests. CLadder (Jin et al., 2023) covers this hierarchy and serves as a holistic causal inference benchmark, requiring models to interpret natural language scenarios, identify query types, and formalize causal expressions, in addition to performing the derivation and calculation needed to answer the query. In our setting, we focus on the derivation and calculation, which require more steps on larger and more interconnected graphs, making them useful for probing how LLMs fine-tuned with RLVR perform as required number of reasoning steps increases. We therefore construct our own dataset, RLCausal, whose questions are about fully specified causal graphs instead of natural language scenarios. We also vary the graph structure, extending beyond CLadder's manually curated topologies to larger, randomly generated 10-node graphs.

Our task is illustrated in fig. 1 top. Concretely, the input $x$ contains a description of a causal graph and a query. For RLVR, the LLM outputs intermediate reasoning steps followed by a probability distribution $\hat{p}$ as the final answer.[1] For SFT, the LLM directly outputs the distribution $\hat{p}$.[2] We define processes for sampling instances of our task, and compute for each instance a reference answer $p^\star$ via variable elimination (Zhang & Poole, 1994).

We then run RLVR and SFT fine-tuning experiments starting from a representative LLM family, Qwen2.5-Instruct (Qwen et al., 2025), varying *model size* and *the level of query* seen during training. Our RLVR training uses variants of GRPO (Shao et al., 2024) and DAPO (Yu et al., 2025b), and our SFT baseline is trained to maximize the probability of $p^\star$ conditioned on task input $x$.

---

[1]The reader would be correct to point out that the causal ladder, contrasts the different *knowledge* required to answer each level of questions (Bareinboim et al., 2022), but our setting with full SCM parametrization as input, eliminates this difference. However, queries from each of levels still need different modes of reasoning— abduction for association, deduction for intervention, and abduction followed by deduction for counterfactual. We discuss in section 3 how our setting affects the difficulty ordering of the three levels.

[2]Additional ablations studying SFT with rejection-sampled reasoning chains are included in appendix D.5.

We present the following findings from our experiments and analysis:

1. **Within- and Across-level Generalization** When trained and evaluated on the same query level, RLVR achieves stronger generalization than SFT on association and intervention queries for models $\geq$ 7B, but under-performs on 3B (for all levels) and counterfactual level (for all sizes); When measuring generalization to a different query level from training, RLVR outperforms SFT on sizes $\geq$ 7B (figs. 3 and 4). RLVR is often more precise than SFT and better on more complex queries (fig. 6).

2. **Scaling and LLM's Reasoning Prior** We trace the effectiveness of RLVR partly back to the strength of the LLM reasoning competence prior to fine-tuning. Scaling up the size of the LLM improves reasoning significantly: 3B models fail to reason before and after RLVR, while an 32B model prompted to reason *zero-shot* beats a 32B model that is *fine-tuned* but predicts the answer directly (fig. 4 bottom).

3. **RLVR improves marginalization strategy and reduces derivation and calculation errors.** Overall, when there is sufficient initial reasoning competence, RLVR shifts the marginalization strategy of LLMs towards incremental marginalization (fig. 5 top), reduces abstract probability/causal derivation errors (e.g. dropping dependencies, confusing intervention with observation) (fig. 5 bottom) as well as calculation errors (fig. 27).

Overall, our findings contribute to the understanding of RLVR's generalization behavior as well as its effectiveness on enhancing LLMs' reasoning capabilities on formal causal reasoning tasks.

## 2 METHOD

We are interested in studying the limits of generalization of RLVR using the task of probabilistic inference in causal graphical models. In this section, we discuss the task definition (section 2.1), training objectives (section 2.2), and the main factors we will vary in our experiments (section 2.3).

### 2.1 TASK DEFINITION

Please refer to fig. 1 (top) for an illustration of the task input and output.

**Input** We use Qwen2.5-Instruct (Qwen et al., 2025) as our base model. The input $x$ for our task consists of a system message $x_{\text{sys}}$ containing short instructions for task and format (fig. 8, appendix) and a user message $x_{\text{user}}$ describing a concrete task instance (fig. 13, appendix), including a description of the causal graphical model and a query. The description of the causal graph includes variable definitions, the graph structure, and mechanism parametrizations.

**Output** For RLVR, the output $y$ is a reasoning chain followed by a probability distribution. For SFT, the output $y$ is directly a probability distribution. See fig. 1 (top) for an illustration of the task for RLVR. We perform extraction of the answer $\hat{p}_y$ from $y$ using regular expression. Each training instance $x$ comes with a reference answer $p_x^\star$.

### 2.2 TRAINING OBJECTIVES

**Reinforcement Learning with Verifiable Rewards** We optimize the RL objective $\mathbb{E}_{x \sim T} \mathbb{E}_{y \sim p_\theta(x)} [r(y)]$, where $T$ is the distribution over training instances. Following typical RLVR setups (DeepSeek-AI et al., 2025), we use a combination of format and accuracy reward, specifically $r(y) = 0.8 \cdot r_{\text{ans}}(\hat{p}_y, p_x^\star) + 0.2 \cdot r_{\text{format}}(y)$, where

$$r_{\text{ans}}(p, q) = \mathbf{1}[D(p, q) < t] \quad r_{\text{format}}(y) = 0.5 \cdot \mathbf{1}[\hat{p}_y \text{ extractable}] + 0.5 \cdot \mathbf{1}[\hat{p}_y \text{ length correct}] \quad (1)$$

and $D(p, q) := \frac{1}{2} \int_x |p(x) - q(x)| \, dx$ is the total variation distance. We round $\hat{p}_y$ and $p_x^\star$ to the nearest two decimal points and use $t = 0.01$.

**Supervised Fine-tuning** For our supervised fine-tuning baseline, we directly maximize the conditional likelihood of the reference answer $y_x^\star$ for input $x$, $\mathbb{E}_{x \sim D} \log p_\theta(y_x^\star \mid x)$.

---

[3] We write intervention queries as $p(\mathbf{v}_{i(\mathbf{v}_j = c)})$ for consistency in notation with the counterfactual queries. It is equivalent to $p(\mathbf{v}_i \mid do(\mathbf{v}_j = c))$ for readers more familiar with the *do* notation (Pearl, 2009).

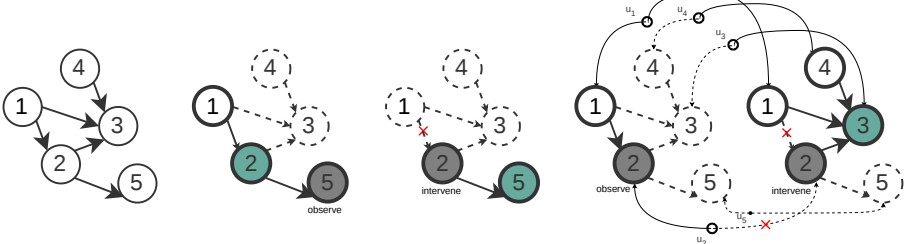

Figure 2: Illustration of graph modifications corresponding to each query level and its relevant (solid) and irrelevant subgraph (dashed). $\times$ denotes dependencies removed due to an intervention. Relevant nodes are defined as ancestors of either the observation or the query variable, after graph modifications are performed to account for any interventions. Left: original graph. Mid-left: 3 relevant nodes for association query $p(\mathrm{v}_2 \mid \mathrm{v}_5 = v_5)$. Mid-right: 2 relevant nodes for intervention query $p(\mathrm{v}_{5(\mathrm{v}_2=c)})$. Right: 10 relevant nodes for counterfactual query $p(\mathrm{v}_{3(\mathrm{v}_2=c)} \mid \mathrm{v}_2 = v_2)$.

## 2.3 STUDY DESIGN

**Data Setup** We stratify the task instances across two axes of difficulty: first, by the query's level (association, intervention, or counterfactual), and second, by the complexity of the query as measured by its relevant subgraph.[4] We vary the source of the training data between individual levels to measure across-level generalization, and breakdown within-level generalization by complexity.

**Fine-tuning Setup** We vary the model size between 3B, 7B, and 32B within the Qwen2.5-Instruct family, using variants of GRPO (Shao et al., 2024) and DAPO (Yu et al., 2025b) for RL, and maximum likelihood of $p^\star$ conditioned on $x$ for SFT.

## 3 DATA GENERATION

In fig. 1 (bottom left) we provide a diagram for the generative process of synthesizing data for our task. We first describe the objects that we sample, then the step by step process for sampling them.

**Structural Causal Models** We use structural causal models (SCMs) (Pearl et al., 2016) with binary variables,[5] no cycles, and independent noise variables as our causal graphical model family. Such a structural causal model $M = (G, \mathbb{F}, \mathbb{Q})$ is defined by a DAG $G = (V, E)$, where each node $i \in V$ is associated with a variable $\mathrm{v}_i$ and a *deterministic* function $f_i \in \mathbb{F}$ that defines its relationship with its parents $\mathrm{pa}(\mathrm{v}_i)$ and a noise variable $u_i$ following distribution $q_i \in \mathbb{Q}$. This gives

$$v_i := f_i(\mathrm{pa}(\mathrm{v}_i), u_i), u_i \sim q_i \tag{2}$$

which induces conditional distributions $p(\mathrm{v}_i \mid \mathrm{pa}(\mathrm{v}_i))$ for all variables $\mathrm{v}_i$.

**Queries** fig. 2 illustrates queries of each level and their graph modifications. Following notation in Pearl et al. (2016), an *association level* query $p(\mathrm{v}_i \mid \mathrm{v}_j = v_j)$ concerns statistical dependence, asking about the distribution of $\mathrm{v}_i$ given an observed value $\mathrm{v}_j = v_j$. An *intervention level* query $p(\mathrm{v}_{i(\mathrm{v}_j=c)})$ concerns causal effects, asking about the distribution of $\mathrm{v}_i$ under an external intervention that sets $\mathrm{v}_j$ to $c$. A *counterfactual level* query $p(\mathrm{v}_{i(\mathrm{v}_j=c)} \mid \mathrm{v}_k = v_k)$ concerns hypothetical alternatives, asking about the distribution of $\mathrm{v}_i$ had $\mathrm{v}_j$ been set to $c$, in a world where we in fact observed $\mathrm{v}_k = v_k$.[6]

**D1: Graph Sampler** The first step of sampling a SCM is sampling a DAG $g$. Given the desired size $N$, we adopt Lampinen et al. (2023)'s workflow to generate a graph, which first samples a random number of independent nodes between 1 and $N$. Additional nodes are then introduced iteratively until we reach $N$ total nodes. Each added node has either one or two parents chosen uniformly from the existing nodes. Finally, the nodes are renamed with a random permutation.

---

[4]Details on this relevant subgraph metric are in section 3.

[5]We choose binary variables for speed of computing the ground-truth solution—exact inference in graphical models is NP-hard in general, and slows down significantly in practice with cardinality and graph size.

[6]The observations/interventions in our data are always on a single variable for simplicity, and LLMs already struggle in this simple setting. Future work could explore vector-valued observations and interventions.

**D2: Mechanism Sampler**    For each $v_i$ and for each joint assignment $\mathbf{v}$ to its parents $pa(v_i)$, we sample a binary distribution $q_{\mathbf{v}}$ uniformly from the simplex. We then define one noise variable $u_i^{\mathbf{v}} \sim q_{\mathbf{v}}$ per each $\mathbf{v}$, and define the mechanism $f_i$ to simply select one particular noise variable's value to take on based on $\mathbf{v}$, namely $v_i = f_i(\mathbf{v}, \boldsymbol{u}_i) = u_i^{\mathbf{v}}$. This simple mechanism directly maps the noise distributions $q_{\mathbf{v}}$ onto rows of the conditional probability table $p(v_i \mid pa(v_i))$.

**D3: Query Sampler**    Given a SCM, we then sample queries for a chosen level. Association level queries contain one observation, intervention level one intervention, and counterfactual level one of each (fig. 2). We sample the target variable $v_i$ uniformly. For association and intervention level queries, we also uniformly sample a variable $v_j$ to condition on or intervene on, respectively. For the counterfactual query, we sample the intervention variable $v_j$ first, and then sample the observation $v_k$ from its descendants. Observations are drawn from the SCM, while interventions uniform $\{0, 1\}$.

**D4: Solver**    Given the full specification of a SCM $M = (G, \mathbb{F}, \mathbb{Q})$ and a query $q$ of association, intervention or counterfactual level, we reduce it to exact inference of some $q'$ in a *possibly modified* SCM $M' = (G', \mathbb{F}', \mathbb{Q})$. We then use variable elimination (Zhang & Poole, 1994) to compute the answer. The modified graphs are illustrated in fig. 2, with additional details in appendix A.1.

**Difficulty Metric**    We first stratify queries by their *level*, as they represent different modes of reasoning. In our setting where we provide the fully parametrized SCM as input, association queries of the form $p(v_i \mid v_j = v_j)$ requires abduction (summing out ancestors in the *posterior*), intervention queries of the form $p(v_{i(v_j=c)})$ requires deduction (sum out ancestors after *fixing* $v_j$), and counterfactual requires abduction followed by deduction (infer *posterior* of noise variables, then sum out noise variables and ancestors in an alternative world after *fixing* $v_j$). This changes the difficulty ordering from the usual association < intervention in the causal ladder to association > intervention in our setting, since computing posterior given $v_j$ usually requires more work than fixing $v_j$ at $c$.[7]

Within each level, we also measure difficulty by $|V_{\text{rel}}|$, the complexity of the query, defined as the size (number of nodes) of the subgraph relevant to the query. Relevant nodes are ancestors of either the observed variable, or the query variable, in the modified graph $G'$. Factors on irrelevant nodes in $V' \setminus V_{\text{rel}}$ sums out to 1 during variable elimination and can be ignored. See fig. 2 for examples.

## 4    EXPERIMENTS

### 4.1    EXPERIMENT SETUP

**Dataset Construction**    For each level in {association, intervention, counterfactual}, we generate a training, development, and test set, consisting of 8000, 2000, and 8000 examples respectively. Each example is a query over a parametrized causal graph over 10 binary variables. We ensure that the SCMs in training, development and test sets are disjoint. Refer to appendix A.2 for more details.

**$|V_{\text{rel}}|$ Distribution**    See fig. 7 (appendix) for histograms of query complexity metric $|V_{\text{rel}}|$. When presenting results, we group examples by ranges of $|V_{\text{rel}}|$ with the following cutoffs: 1-3 (small), 4-6 (medium), 7-10 (large) for association, 1-2 (small), 3-4 (medium), 5-10 (large) for intervention, and 1-7 (small), 8-15 (medium), 16-30 (large) for counterfactual. The complexities are meant to be used within-level and is generally *not* comparable across levels.

**Metrics**    Given a input $x$ for a language model, its reference solution $p_x^\star$, and a LLM output $y$, we measure its correctness by the following metric based on total variation distance and a format requirement. Let $\hat{p}_y$ be a solution extracted from $y$. Let $\hat{p}_y$ and $p_x^\star$ be rounded to 0.01,

$$\text{CORRECT}_t(x, y) := \begin{cases} 0 & \text{if format error, failed to extract } \hat{p}_y \\ 1 & \text{if } D(\hat{p}_y, p_y^\star) \le t \end{cases} \tag{3}$$

where $D(p, q) := \frac{1}{2} \int_x |p(x) - q(x)| \, dx$ is the total variation distance, and $t = 0.01$.

**Filtering**    Due to rounding, on many instances the intervention and observation may not result in a measurable change in the marginal distribution of the query variable, making the instance insensitive to faithful reasoning. Therefore, we filter out such examples in our main analysis. We also include the unfiltered results in appendix D.

---

[7]If we had also included intervention queries with *conditions*, then it would require graph modification followed by abduction, and the difficulty ordering would be reverted back to the usual association < intervention.

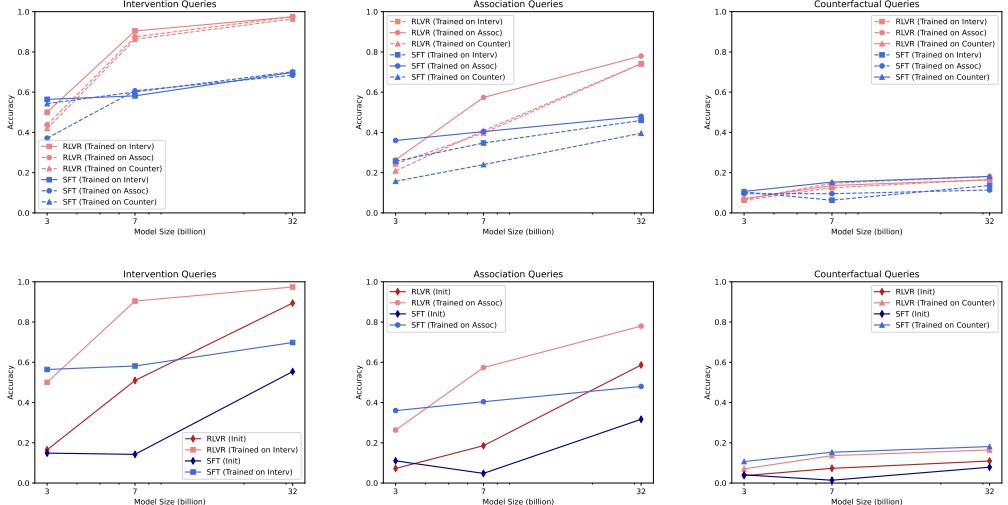

Figure 4: Top: Accuracy (y-axis) vs. LLM size (x-axis) when evaluated on intervention (left), association (middle), and counterfactual (right) queries. Red curves correspond to RLVR, blue curves correspond to SFT. Solid (−) curves are LLMs fine-tuned on the *same* level as evaluation, dashed (−−) curves are trained on a *different* level from evaluation. Bottom: Reasoning (RLVR) vs non-reasoning (SFT) strategies, before and after fine-tuning. As scale increases, both reasoning and non-reasoning prior improve, though the reasoning prior benefits more from scaling.

**Fine-tuning And Inference Setup** **For RLVR**, We use GRPO with the token-level normalization, and DAPO without the overlong buffer for simplicity. We use a batch size of 8, with 32 roll-outs per example, a learning rate of $10^{-6}$, and other hyperparameters default from DAPO implemented in the VERL library (Sheng et al., 2024). We train for 7.5k steps for 3B and 7B models, and 2.5k steps for 32B models, as they are roughly a third slower to train compared to 7B. **For SFT**, We train using maximum likelihood on $p^\star$ for 5k steps, with learning rate $10^{-6}$ and pick best checkpoint (saved every 200 step) by picking best loss on development set. **For inference**, we decode at temperature 0. Additional details on hyper-parameters are included in appendix B.1.

## 4.2 MAIN RESULTS

We focus our discussion and analysis on GRPO as the representative RLVR algorithm in the main text, as it is simple, widely used, and its results are not significantly different from DAPO in our experiments. DAPO results are included in appendix D.

In fig. 4, we compare the accuracy of LLMs fine-tuned via RLVR and SFT on the three query types—intervention, association, and counterfactual. We vary the model size between 3B and 32B, and we vary the query type that the model was fine-tuned on.

**Within-level Generalization: RLVR outperforms SFT on only a subset of (model size, query type) configurations.** In fig. 3 left, we show the size and query type configurations for which RLVR outperforms SFT, when trained and evaluated on the same query type —RLVR significantly outperforms SFT on intervention and association queries, for sizes ≥ 7B.

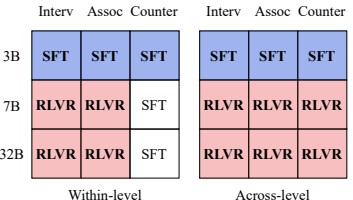

Figure 3: The algorithm with higher within-/across-level accuracy for different sizes and query types. Significant cells (paired-perm test at $p < 0.05$) bolded and colored.

**Across-level Generalization: RLVR outperforms SFT beyond 3B models. Larger LLMs perform better across levels for both RLVR and SFT.** In fig. 3 right, we see when evaluating on different level from training, RLVR outperforms SFT on models ≥ 7B. In fig. 4 we find that for both SFT and RLVR, the performance gap between LLMs fine-tuned on in- and out-of-level queries generally decreases as model scale increases, suggesting better cross-level generalization with scaling.

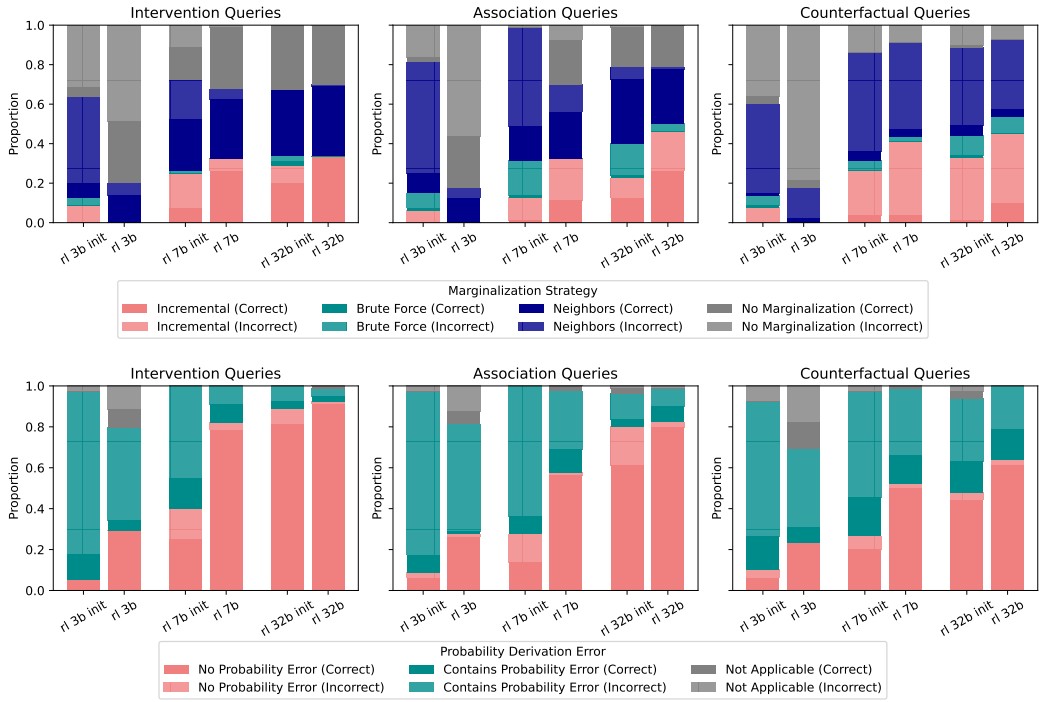

Figure 5: LLM judge (o4-mini) analysis of the marginalization strategy (top) and the existence of derivation errors (bottom) before and after RLVR. Derivation errors and marginalization strategies are annotated on (the same) 80 samples per level. Judge prompts (including category definitions) are included in fig. 11. Example traces of marginalization strategy are included in figs. 14 to 17.

## 4.3 ANALYSIS

**Analysis-I: RLVR is ineffective when the reasoning capability of the base model is too poor prior to fine-tuning.** Why is RLVR ineffective on 3B and counterfactual queries, as seen in fig. 3?

*3B models attempt explicit marginalization before fine-tuning, but succeed rarely; After fine-tuning it outputs answer directly without explicit marginalization.* We reviewed a subset of reasoning traces across levels and sizes, before and after RLVR fine-tuning. All models attempt explicit marginalization before fine-tuning, but only models $\geq$ 7B continue attempting explicit marginalization after fine-tuning. Our analysis of reasoning traces in fig. 5 show that at 3B, traces that attempt to marginalize step by step (incremental, brute force, and neighbors) are rarely correct—a possible explanation for their regression to directly predicting the answer after fine-tuning. See fig. 18 for an example trace from 3B model with errors annotated, and see *Analysis-III* for a more detailed discussion.

*On counterfactual level queries, models did not attempt to build twin-networks or perform inference over exogenous variables, both before and after RLVR.* We reviewed a subset reasoning traces from models of different sizes, but did not observe any attempts to create a twin-network-graph. We conducted an oracle experiment where additional hints on solving the counterfactual query via twin network graph is provided in the system prompt (fig. 9, appendix). However, its accuracy is not very different from the original prompt without hints (fig. 19). This result is in contrast to more positive findings in previous evaluations of LLMs' counterfactual reasoning in the commonsense settings (Kıcıman et al., 2024) or formal settings with continuous mechanisms(Tu et al., 2024). This contrast may be partly due to our strict metric and more numerically challenging task.

Overall, these findings suggest that RLVR is sensitive to the LLM's reasoning success rate prior to fine-tuning, echoing the cold start problem (DeepSeek-AI et al., 2025) in RLVR post-training. Since we start from Qwen2.5-Instruct (Qwen et al., 2025), which is already instruction-tuned and has some reasoning-tuning too, the limitations seen here is likely more specific to the causal domain.

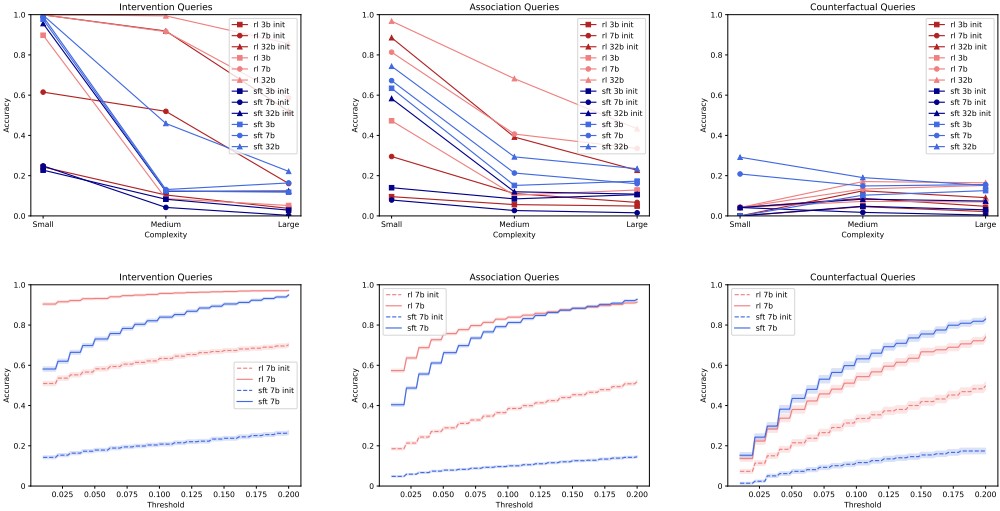

Figure 6: Top: Within each query level, accuracy vs. query complexity $V_{rel}$ increases. Note that complexities are not comparable across levels. Bottom: Accuracy as the threshold for correctness $t \in (0.01, 0.2]$ is relaxed for 7B models. $x$-axis plots threshold for accuracy $t$ (the lower the stricter), and $y$-axis plots accuracy at $t$. Across all levels (left to right), we see that RL models are often more precise than SFT models. See figs. 22 to 24 for the same plot but for all query levels and model sizes. The observable staircase pattern is due to rounding to two digits.

**Analysis-II: LLM's success rate prior to fine-tuning improves with scale and especially with reasoning. The prior is a major source of RLVR's effectiveness.** In fig. 4 (bottom), we show the initial accuracy of LLMs prompted to reason (used in RLVR) versus LLMs prompted to directly predict the answer (used in SFT). Across all levels, both the reasoning prompt and the direct prediction prompt's accuracy prior to fine-tuning increases substantially with scale, but reasoning benefits more from scaling. Furthermore, the reasoning prompt prior to fine-tuning achieves a higher accuracy than non-reasoning prompt across the board. Notably, on intervention and association queries, 32B models fine-tuned with SFT still under-performs zero-shot reasoning (fig. 4 bottom). This demonstrates the substantial benefit of the reasoning prior, which we next show RLVR improves on.

**Analysis-III: RLVR favors incremental marginalization strategy and reduces abstract reasoning errors and calculation errors.** Answering queries from our causal inference task requires abstract reasoning about probability and causality (e.g. marginalization and graph modification) as well as low-level calculations (e.g. substitution of values and arithmetic). See example in fig. 10.

To *qualitatively* understand LLMs' performance on these sub-tasks, we manually examined a subset of reasoning traces across all levels. We found that incorrect answers are often due to errors in abstract reasoning about probability or causality (e.g. falsely assuming independence, dropping terms in a marginalization formula, or treating a counterfactual query as an intervention / association query), and can sometimes also be due to low-level calculation errors (e.g. copying wrong probability values from the problem statement or making numerical mistakes in addition / multiplication).

To *quantitatively* understand LLMs' improvement on these sub-tasks after RLVR, we analyzed the reasoning traces from our RLVR models with o4-mini. We focus here on presenting analyses of abstract reasoning errors and marginalization strategies (and leave low-level calculations to appendix D.7)—abstract reasoning errors were salient in our qualitative analysis, and marginalization influences the high-level solution strategy.[8]

*Abstract Reasoning Errors*: In fig. 11 (bottom) we show our prompt to o4-mini for analyzing abstract reasoning errors in an LLM's reasoning chains. The prompt focuses on two sub-types of

---

[8]The first author validated the prompts for analyzing reasoning traces with 10 samples per category of marginalization strategy / reasoning error found 25/30 agreement with o4-mini on reasoning error detection and 37/40 agreement on marginalization strategy detection.

errors, one where some probability identity was used incorrectly, such as errors in the formula for the chain rule/Bayes rule, and another where some false assumption was made, such as assuming unwarranted independences when computing joint probabilities or falsely treating interventions as observations. In fig. 5 (bottom) we show the comparison of the rate of abstract reasoning errors detected by o4-mini (indicated with color) in reasoning chains before and after RLVR, and whether the final answer was incorrect (indicated with shading). Overall, we see that larger models make much less probability derivation errors, RLVR fine-tuning significantly reduces probability derivation errors, and the existence of such errors correlate highly with incorrectness of the final answer.

*Marginalization:* In fig. 11 top we show our prompt for o4-mini to annotate the marginalization strategy used in a reasoning chain. We identified three distinct marginalization strategies during our qualitative analysis: *brute-force* summation, where the solution involves large summation formulas (e.g. $p(v_4, v_5 = 1) = \sum_{v_1, v_2, v_3} p(v_1, v_2, v_3, v_4, v_5 = 1)$), *incremental* marginalization, which sums out one or two variables at a time (e.g. first marginalize out $v_3, v_4$ then $v_2$), and *no-marginalization*, which can be due to a trivial query like $p(v_{i_{(v_i=1)}})$, or due to LLM predicting a numeric answer directly (often incorrectly) without calculations. To address ambiguity between "brute-force" and "incremental marginalization" on the base-case of summing over the immediate neighbors (as it can be seen as a special case of both), we add a fourth category "neighbors" to avoid ambiguity during o4-mini's analysis. See figs. 14 to 17 for an example of each category.

In fig. 5 (top), we see that for **7b and 32b models**, RLVR shifts the marginalization strategy towards incremental marginalization, and we found that this effect is especially strong on more difficult queries (see fig. 21 top)—those with a larger number of relevant nodes to sum over. A possible explanation for this shift could be that brute-force summation often introduces large summations with terms that are long products of conditional probabilities, which can give rise to more chances of errors, causing incorrect answers (green shaded), and resulting in RLVR learning to avoid it. In fig. 5 (top), we also see that for **3b models**, instead of learning to marginalize correctly, they learned to avoid marginalization after RLVR (grey), frequently predicting the answer without any calculations (see fig. 17 for an example). The high rate of failure (indicated by shading) of marginalization-based solutions (incremental, brute-force, and neighbors) prior to training is a possible explanation for why 3b LLMs avoided it after RLVR.

**Analysis-IV: How does RLVR models behave differently from SFT models?** In fig. 6 (top), we show that within each query type, on 7B and 32B LLMs, SFT tend to perform better on the less complex queries of that type, while RLVR performs better on more complex queries of that type. In fig. 6 (bottom), we plot the proportion of correct answers as we relax the criterion of correctness from exact match to within 0.2 in total variation distance. We show that LLMs fine-tuned with RLVR are more precise, while SFT models often is only able to get the solution approximately correct. Both RLVR and SFT significantly improve the precision of the LLMs relative to base.

## 5 RELATED WORK

**RLVR for LLM Post-Training** Reinforcement learning is becoming a common step of LLM post-training. Early works use RL to align LLMs using human preference data (Ouyang et al., 2022). Reinforcement learning with verifiable rewards (RLVR) (Lambert et al., 2025; DeepSeek-AI et al., 2025) teach reasoning on domains where reward (often a combination of correctness and format) can be verified automatically. This led to significant progress in domains such as math problem solving theorem proving, and code generation (DeepSeek-AI et al., 2025; Ren et al., 2025; Liu & Zhang, 2025). In this paper, we attempt to understand the successes and limitations of RLVR's generalization, using a controlled family of causal reasoning tasks.

**Understanding RLVR's Generalization** Several recent studies explored the respective contribution of SFT and RL to the *generalization* of the resulting LLM (Chu et al., 2025; Chen et al., 2025; Swamy et al., 2025), with evidence that SFT is useful for warm-starting, while RL can significantly improve generalization. Another line of work investigates how SFT and RL should be ordered or combined to achieve the best generalization result (Liu et al., 2025b; Qiu et al., 2025). Swamy et al. (2025) and Qin et al. (2025) study the sources of RLVR's effectiveness and where it improves models, and Yue et al. (2025); Sun et al. (2025); Liu et al. (2025a) and Wu & Choi (2025) study changes on the generalization boundary of LLMs fine-tuned with RLVR. Similar to the earlier works, we investigate generalization of RLVR and SFT, and similar to the latter works, we focus on under-

standing the limits and the sources of RL's effectiveness itself. Different from these works, we focus on the causal reasoning domain, a less explored but important area that remains challenging.

**LLM for Causal Reasoning** LLMs' understanding of causality has been receiving increasing interest (Yu et al., 2025a), as LLMs make their way into domains like medicine and law where causal reasoning is essential. One line of work investigates the real world causal knowledge and reasoning of pretrained LLMs (Kıcıman et al., 2024; Zečević et al., 2023; Chi et al., 2024). These benchmarks are valuable for assessing whether LLMs encode plausible causal facts about the world, but they are less suited for probing RLVR, which focuses more on step-by-step reasoning over such facts to draw new conclusions, a separate skill from memorization of the facts or conclusions. A line of datasets that focus more on formal causal reasoning capabilities include CLadder, Corr2Cause, and Carl-gt (Jin et al., 2023; 2024; Tu et al., 2024). Among these more formal benchmarks, the closest to ours is CLadder (Jin et al., 2023), whose questions and answers are algorithmically generated by running a causal-inference engine over structured graphical models and expressed as natural-language scenarios spanning all rungs of the causal ladder. Similar to CLadder, our benchmark is built on synthetic casual graphical models, focusing more on reasoning over memorization. Different from CLadder, our benchmark isolates causal reasoning from natural language understanding by providing a full specification of the abstract causal graphic in the question rather than converting it to a natural language scenario. We also introduce substantially more structural diversity by using larger graphs (10 nodes) and a wider range of randomly generated graph topologies (80 distinct for training and 200 distinct for dev/test). These two features make our dataset a focused and challenging dataset suitable for studying within-level and cross-level generalization of RLVR.

## 6 CONCLUSION AND FUTURE DIRECTIONS

In this paper, we investigated the generalization behavior of RLVR and SFT using probabilistic inference in causal graphical models as a testbed. Our main findings are that RLVR is less effective when the task is too challenging for the model prior to fine-tuning, but when the LLM has a basic level of successful reasoning rate on a problem level, RLVR significantly improves generalization, by fixing domain-specific reasoning errors and boosting systematic marginalization strategies, outperforming SFT, and especially on more complex queries. We also find that the LLM's prior (both reasoning and direct prediction) scales with model size, and scaling gains are stronger with reasoning.

Our findings add to an emerging line of investigation into RLVR's generalization (Chu et al., 2025; Chen et al., 2025; Swamy et al., 2025). A future direction is to explore more deeply the roles of execution quality and strategy quality in reasoning and RLVR (e.g. see recent works Qin et al., 2025; Sinha et al., 2025). This may provide us more insights into the mechanism of RLVR's generalization.

Our findings also add to an increasing body of work studying LLMs for causal reasoning (Kıcıman et al., 2024; Jin et al., 2023; 2024). Their effectiveness on some commonsense counterfactual causal reasoning settings (Kıcıman et al., 2024) contrasts with our results on more challenging formal counterfactual reasoning problems. A possible future direction is to understand this gap better.

Recent work has started building domain-specific task suites for specializing LLMs to solve complex tasks in real-world science and engineering domains via RLVR fine-tuning. For example, Biomni-R0 (Biomni & Sky RL, 2025) uses RLVR to fine-tune LLMs on a range of biomedical tasks that requires step-by-step reasoning, and ether0 (Narayanan et al., 2025) does so similarly for chemistry. Beyond supporting our analysis of RLVR generalization, our dataset can be used to train and probe subskills (e.g. organizing marginalization of latent variables, applying probability identities, performing arithmetic) that are useful to solving more practical causal and probabilistic problems, even though our problems are abstracted away from real-world scenarios. Our data generation process can also be configured to scale up the number of problems as well as their difficulty, by increasing graph size and the cardinality of each variable as appropriate.

ACKNOWLEDGEMENTS

This work was supported by a research gift to the last author by Adobe Research. This work is also supported by NSF Grant No. CHE2505932, an Amazon AICE Award, gift funding from AI2, and a grant from Coefficient Giving. The views and opinions of authors expressed herein do not necessarily state or reflect those of the United States Government or any agency thereof. We thank the anonymous ICLR reviewers and meta-reviewers for their constructive feedback.

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

## A  ADDITIONAL DETAILS ON DATA

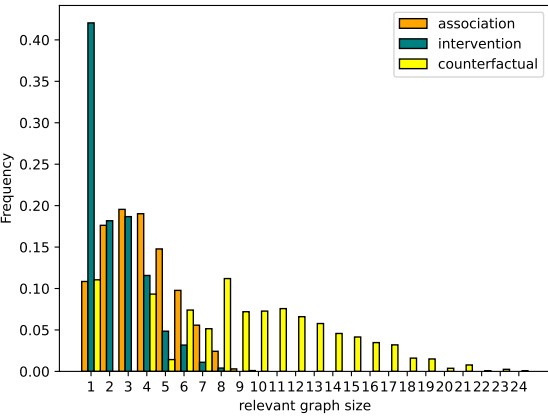

Figure 7: Distribution of query difficulty within each level, as measured by size of relevant subgraph defined in section 3.

### A.1  GRAPH MODIFICATIONS FOR EACH LEVEL

For an association level $q$ of the form $p(\mathrm{v}_i \mid \mathrm{v}_j = v_j)$, we keep the same graph and query: $M' = M$ and $q' = q$.

For an intervention level query of the form $p(\mathrm{v}_{i(\mathrm{v}_j = c)})$, we modify the graph and query. We make $q' = p(\mathrm{v}_i)$, and $M'$ a modification of $M$ where we replace the mechanism for $\mathrm{v}_j$ with a constant function $f_j = c$, resulting with $\mathbb{F}' = \mathbb{F}_{-j} \cup \{f_j = c\}$. We also remove any incoming edges to $\mathrm{v}_j$, resulting with $G' = (V, E \setminus \{(k \to j) \mid k \in V\})$.

For a counterfactual query $p(\mathrm{v}_{j(\mathrm{v}_i = c)} \mid \mathrm{v}_k = v_k)$, we modify the graph and query. We create $M'$ by first augmenting it with a twin copy of each endogenous node v denoted $\mathrm{v}^{\mathrm{twin}}$. This results with $G' = (V \cup V^{\mathrm{twin}}, E \cup E^{\mathrm{twin}})$. $E^{\mathrm{twin}}$ mirrors $E$ but connects twin copy nodes. The mechanisms for $\mathrm{v}_i^{\mathrm{twin}}$ uses parents in $V^{\mathrm{twin}}$ but share noise $\mathrm{u}_i$ with the original $\mathrm{v}_i$. Then the same graph surgery and mechanism replacement is applied to the *twin* copy $\mathrm{v}_j^{\mathrm{twin}}$ to account for the intervention. We then define $q' = p(\mathrm{v}_j^{twin} \mid \mathrm{v}_k = v_k)$ in $M'$. See (Shpitser & Pearl, 2012) for more discussions on twin network graph that handles two possible worlds (enough for our setting) and its generalization *parallel worlds graph* that handles more.

All the modified queries $q'$ are association level queries, so we can use standard graphical model inference, e.g. variable elimination Zhang & Poole (1994), in $M'$ to compute the answer.

See fig. 2 for example graph modifications of each level.

### A.2  ADDITIONAL DATASET DETAILS

The training questions is sampled on 80 different graphical models, with 100 queries sampled per model. The development and test sets are sampled each on 200 different graphical models, with 10/40 queries per model for dev/test.

## B  ADDITIONAL DETAILS OF EXPERIMENT SETUP

### B.1  ADDITIONAL HYPERPARAMETERS

**RLVR**  For GRPO variant, we train 3B and 7B models for 7.5k steps, 32B models for 2.5k steps. For DAPO variant, we train 3B and 7B models for 2.5k steps, and 32B models for 850 steps. We use the final checkpoint for all. These step heuristics chosen to roughly control the amount of gpu

hours spent on each run (32B is about 3x slower than 7B, and DAPO is roughly another 3x slower in terms of time per step due to filtering). For 3B models with GRPO and DAPO, the dev performance already plateaued early due to regression to direct prediction, so we did not train further beyond 7.5k/2.5k steps, respectively.

Additional hyper-parameters for our GRPO variant include a 0 coefficient on the KL term, a PPO clip ratio low of 0.2, high of 0.28, and c of 10, and token level averaging when computing the advantage function, following enhancements described in DAPO (aside from dynamic sampling) Yu et al. (2025b) and its implementation in VeRL (Sheng et al., 2024) . We use Adam optimizer (Kingma & Ba, 2017) with default parameters, weight decay of 0.1 and no warmup steps.

For DAPO runs, we focus on its curriculum aspect, disabling the overlong buffer, and setting a stricter filtering of accuracy between $(0.09, 0.91)$ which corresponds to more than 3 out of 32 rollouts being correct and more than 3 out of 32 rollouts being incorrect. We chose this filtering range in hopes of seeing stronger effects with filtering compared to $(0, 1)$ range originally used in DAPO (Yu et al., 2025b). However, our results showed no significant difference compared to GRPO. See results in appendix D with rows labeled with "cur" for curriculum.

## C  PROMPTS AND REASONING OUTPUTS

**System Prompts**    In fig. 8 we include the system prompt for RLVR and SFT. In fig. 9 we include the system prompt with hint for counterfactual level.

**User Prompt**    In fig. 13, we show a full example of task input.

**Reasoning Traces**    In figs. 14 to 17 we show reasoning traces of the four strategy categories "Incremental", "Brute Force", 'Neighbors", and "No Marginalization".

**System Prompt RLVR**

You are an expert on graphical models and causal inference. You task is to compute probability queries over a structural causal model.

Format your solution as follows:

THOUGHT PROCESS
All intermediate steps of analysis, reasoning, and computation.

ANSWER
Only state your final answer to the query, via a list of numbers such as [0.1, 0.7, 0.2] or [0.1, 0.9]. Round you answer to the nearest two digits.

---

**System Prompt SFT**

You are an expert on graphical models and causal inference. You task is to compute probability queries over a structural causal model.

Format your solution as follows:

Only state your final answer to the query, via a list of numbers such as [0.1, 0.7, 0.2] or [0.1, 0.9]. Round you answer to the nearest two digits.

Figure 8: System Prompt for RLVR and SFT.

**System Prompt RLVR Hint For Counterfactual**

You are an expert on graphical models and causal inference. Your task is to compute probability queries over a structural causal model (SCM).

## Model Specification
You will be given the specification of an SCM:
* The graph structure will be provided.
* The parametrization of mechanisms will be specified with conditional probability tables (CPTs).

Each endogenous variable v_i is associated with a collection of independent exogenous variables-one for each possible joint assignment of its parents-that follows the same conditional distribution P(v_i | parents(v_i)). Given a joint assignment of its parents and the values of the corresponding exogenous variables, the mechanism assigns v_i the value of the appropriate exogenous variable.

## Counterfactual Queries
To compute a counterfactual query:
1. Construct a twin network:
* Copy all endogenous variables.
* For each pair (v_i_original, v_i_copy), they share the same exogenous variables but connect to their respective parent sets (original vs. copied).
2. Interpret the query as a standard probability inference problem:
* Observations are applied to the original variables.
* Interventions are applied to the copied variables.
* The query target is also a copied variable.

## Format
Format your solution as follows:
THOUGHT PROCESS
All intermediate steps of analysis, reasoning, and computation.

ANSWER
Only state your final answer to the query, via a list of numbers such as [0.1, 0.7, 0.2] or [0.1, 0.9]. Round you answer to the nearest two digits.

Figure 9: System Prompt for RLVR with hint.

THOUGHT PROCESS
1. Intervention on $v2$: We need to compute the marginal distribution of $v1$ after setting $v2 = 0$. This means we need to consider the distribution of $v1$ given $v2 = 0$ and sum over all possible values of $v5$ (since $v5$ is a parent of $v1$). (graph modification unnecessary for this case)
2. CPT for $v1$ given $v2 = 0$: The CPT for $v1$ given $v2$ and $v5$ is: (copy probability values)
- $P(v1|v2 = 0, v5 = 0) = [0.83, 0.17]$
- $P(v1|v2 = 0, v5 = 1) = [0.22, 0.78]$
3. Marginalize over $v5$: We need to sum over the probabilities of $v5$ (which is $P(v5) = [0.49, 0.51]$).
4. Compute the marginal distribution: (marginalize over $v_5$)
- $P(v1 = 0|v2 = 0) = P(v1 = 0|v2 = 0, v5 = 0) \cdot P(v5 = 0) + P(v1 = 0|v2 = 0, v5 = 1) \cdot P(v5 = 1)$
- $P(v1 = 1|v2 = 0) = P(v1 = 1|v2 = 0, v5 = 0) \cdot P(v5 = 0) + P(v1 = 1|v2 = 0, v5 = 1) \cdot P(v5 = 1)$
Let's compute these values step by step. (substitution and arithmetic)

$$P(v1 = 0|v2 = 0) = 0.83 \cdot 0.49 + 0.22 \cdot 0.51 = 0.4067 + 0.1112 = 0.5179$$

$$P(v1 = 1|v2 = 0) = 0.17 \cdot 0.49 + 0.78 \cdot 0.51 = 0.0833 + 0.3978 = 0.4811$$

Rounding to two decimal places, we get:

$$P(v1 = 0|v2 = 0) = 0.52 \quad P(v1 = 1|v2 = 0) = 0.48$$

ANSWER
$[0.52, 0.48]$

Figure 10: Illustration of sub-tasks involved in computing $p(\mathrm{v}_{1_{(v_2=0)}})$. $\mathrm{v}_2$ has no parents in this particular graph so $p(\mathrm{v}_{1_{(v_2=0)}}) = p(\mathrm{v}_1 \mid \mathrm{v}_2 = 0))$. This is an example "neighbors" marginalization strategy, performing summations over $\mathrm{v}_1$'s immediate parents.

**LLM Prompt, Marginalization Strategy Categorization**

You are an expert grader that never makes mistakes.

### Input
You will be given a LLM's SOLUTION for a QUESTION that asks for the marginal distribution of some random variable $v_i$ under an intervention, observation, or counterfactual (hypothetical intervention under an observation).

### Analysis
A correct strategy need to marginalize over other relevant variables correctly. You need to determine if the solution strategy is one of the following:
* immediate: attempt to marginalize only over immediate neighbors
* incremental: attempt to marginalize over neighbors as well as other more distant variables, performing marginalization incrementally, often following the graph structure and performing summations locally over one subset of variables at a time for many times.
* brute: attempt to marginalize over neighbors as well as other more distant variables, but does so by explicitly writing out a main formula that sums over the joint probability distribution over ALL relevant variables together (which often involves many terms), instead of summing over smaller subsets many times.
* none: no attempt at explicit marginalization.

### Formatting Response
You need to output one judgement specified below: for that judgement, put any relevant EVIDENCE, which are excerpts from SOLUTION, within an <evidence></evidence> tag, then your EXPLANATION, if any, within the <explanation></explanation> tag, and finally put your judgement in <judgement></judgement> tag.
1. Overall Strategy
Use <evidence_strategy></evidence_strategy>, <explanation_strategy></explanation_strategy>, and <judgement_strategy></judgement_strategy>. Choose judgement between "immediate", "incremental", "brute", "none".

---

**LLM Prompt, Probability Derivation Errors**

You are an expert grader that never makes mistakes.
### Input
You will be given a LLM's SOLUTION for a QUESTION that asks for the marginal distribution of some random variable $v_i$ under an intervention, observation, or counterfactual (hypothetical intervention under an observation).

### Analysis
A correct SOLUTION needs to perform derivations correctly and perform calculations correctly. Your task is to identify any probability derivation errors in the derivation. If you believe there are any errors the precise error location in the solution must be identified.

Probability derivation errors include:
* errors when applying probability identities (e.g. applying chain rule or bayes rule incorrectly, summing over too many or too few variables when marginalizing)
* false assumptions (e.g. ignoring dependencies between variables when performing inference, ignoring observations or interventions)
* ...

Probability derivation errors does NOT include:
* errors in copying CPT values
* numeric errors (e.g. incorrectly performing addition or multiplication)

### Formatting Response
You need to output one judgement specified below: for that judgement, put any relevant EVIDENCE, which are excerpts from SOLUTION, within an <evidence></evidence> tag, then your EXPLANATION, if any, within the <explanation></explanation> tag, and finally put your judgement in <judgement></judgement> tag.

1. Derivation Error
Use <evidence_derivation_error></evidence_derivation_error>, <explanation_derivation_error></explanation_derivation_error>, and <judgement_derivation_error></judgement_derivation_error>.

Choose your judgement from "yes" (solution contains derivations, and contains probability derivation errors), "no" (solution contains derivations, but no probability derivation errors detected), or "n/a" (solution does not contain any derivations).

Figure 11: Prompts for LLM Judge for strategy categorization and detecting probability derivation errors.

**LLM Prompt, Copy Error Detection**

You are an expert grader that never makes mistakes.

### Input
You will be given a LLM's SOLUTION for a QUESTION that asks for the marginal distribution of some random variable $v_i$ under an intervention, observation, or counterfactual (hypothetical intervention under an observation).

### Analysis
A correct SOLUTION needs to perform derivations correctly and perform calculations correctly. Your task is to identify any copy-paste errors on the conditional probability values used in the derivation. If you believe there are any errors the precise error location in the solution must be identified.

### Formatting Response
You need to output one judgement specified below: for that judgement, put any relevant EVIDENCE, which are excerpts from SOLUTION, within an <evidence></evidence> tag, then your EXPLANATION, if any, within the <explanation></explanation> tag, and finally put your judgement in <judgement></judgement> tag.

1. Conditional probability copy-paste Error
Use <evidence_copy_error></evidence_copy_error>, <explanation_copy_error></explanation_copy_error>, and <judgement_copy_error></judgement_copy_error>.

Choose your judgement from "yes" (solution contains derivations, and contains conditional probability copy-paste errors), "no" (solution contains derivations, but no conditional probability copy-paste errors detected), or "n/a" (solution does not contain any derivations).

---

**LLM Prompt, Arithmetic Error Detection**

You are an expert grader that never makes mistakes.

### Input
You will be given a LLM's SOLUTION for a QUESTION that asks for the marginal distribution of some random variable $v_i$ under an intervention, observation, or counterfactual (hypothetical intervention under an observation).

### Analysis
A correct SOLUTION needs to perform derivations correctly and perform calculations correctly. Your task is to identify any arithmetic errors in the derivation (when adding / subtracting / multiplying / dividing numbers). It is not considered an error to perform rounding at intermediate steps. If you believe there are any errors the precise error location in the solution must be identified.

### Formatting Response
You need to output one judgement specified below: for that judgement, put any relevant EVIDENCE, which are excerpts from SOLUTION, within an <evidence></evidence> tag, then your EXPLANATION, if any, within the <explanation></explanation> tag, and finally put your judgement in <judgement></judgement> tag.

1. Arithmetic Error
Use <evidence_arithmetic_error></evidence_arithmetic_error>, <explanation_arithmetic_error></explanation_arithmetic_error>, and <judgement_arithmetic_error></judgement_arithmetic_error>.

Choose your judgement from "yes" (solution contains derivations, and contains arithmetic errors), "no" (solution contains derivations, but no arithmetic errors detected), or "n/a" (solution does not contain any derivations).

Figure 12: Prompts for LLM Judge for detecting copy error and arithmetic error in solution.

**User Prompt All Levels**

Here's a structural causal model over discrete random variables. The Variables are v0, v1, v2, v3, v4, v5, v6, v7, v8, v9. Here are the Values they can take on.

v0 can take values in [0, 1]
v1 can take values in [0, 1]
v2 can take values in [0, 1]
v3 can take values in [0, 1]
v4 can take values in [0, 1]
v5 can take values in [0, 1]
v6 can take values in [0, 1]
v7 can take values in [0, 1]
v8 can take values in [0, 1]
v9 can take values in [0, 1]

Here's the causal directed acyclic graph (DAG):
strict digraph {
v0; v1; v2; v3; v4; v5; v6; v7; v8; v9; v3 → v0; v3 → v5; v4 → v1; v4 → v3; v4 → v7; v4 → v8; v4 → v9;
v8 → v2; v8 → v6; v8 → v7; v9 → v3; }

Here are the causal conditional probability tables (CPT) associated with the DAG:
CPTs for v4:
P(v4) = [0.51, 0.49]

CPTs for v8:
P(v8| v4=0) = [0.02, 0.98]
P(v8| v4=1) = [0.36, 0.64]

CPTs for v7:
P(v7| v8=0,v4=0) = [0.94, 0.06]
P(v7| v8=0,v4=1) = [0.25, 0.75]
P(v7| v8=1,v4=0) = [0.49, 0.51]
P(v7| v8=1,v4=1) = [0.58, 0.42]

CPTs for v2:
P(v2| v8=0) = [0.11, 0.89]
P(v2| v8=1) = [0.97, 0.03]

CPTs for v9:
P(v9| v4=0) = [0.95, 0.05]
P(v9| v4=1) = [0.42, 0.58]

CPTs for v3:
P(v3| v9=0,v4=0) = [0.46, 0.54]
P(v3| v9=0,v4=1) = [0.61, 0.39]
P(v3| v9=1,v4=0) = [0.7, 0.3]
P(v3| v9=1,v4=1) = [0.77, 0.23]

CPTs for v6:
P(v6| v7=0,v8=0) = [0.1, 0.9]
P(v6| v7=0,v8=1) = [0.5, 0.5]
P(v6| v7=1,v8=0) = [0.47, 0.53]
P(v6| v7=1,v8=1) = [0.53, 0.47]

CPTs for v5:
P(v5| v3=0) = [0.38, 0.62]
P(v5| v3=1) = [0.66, 0.34]

CPTs for v0:
P(v0| v3=0) = [0.96, 0.04]
P(v0| v3=1) = [0.86, 0.14]

CPTs for v1:
P(v1| v4=0) = [0.59, 0.41]
P(v1| v4=1) = [0.7, 0.3]

(**Included Only For Counterfactual level**: Furthermore, each variable v is assumed to depend deterministically on its parents pa(v) and a collection of independent exogenous selector variables, one for each possible joint assignment to pa(v), whose marginal distribution is defined to be p(v | pa(v)). Given a particular assignment to pa(v), v takes on the value of the selector variable corresponding to that particular assignment pa(v).)

Here's your Question: {Question}
———
Now start your solution process. Be precise.

---

**Association Question**
What is the marginal distribution of v6 iven it is observed that v8=1?

**Intervention Question**
What is the marginal distribution of v6 given we intervented to set v8 to 1?

**Counterfactual Question**
What is the marginal distribution of v3 given we first observed v6 = 0 and then intervened to set v8 to 1?

Figure 13: Example user prompt $x_{\text{user}}$ containing causal graph and query.

**Incremental Marginalization**

THOUGHT PROCESS
Given the intervention $\text{do}(v1 = 0)$, the distribution of $v1$ is fixed at 0. This means we need to consider the conditional probability tables (CPTs) for the other variables given $v1 = 0$.
The CPT for $v6$ is given by:

$$P(v6|v0, v1 = 0) = \begin{cases} P(v6|v0 = 0, v1 = 0) & \text{if } v0 = 0 \\ P(v6|v0 = 1, v1 = 0) & \text{if } v0 = 1 \end{cases}$$

From the CPTs provided:

$$P(v6|v0 = 0, v1 = 0) = [0.65, 0.35]$$

$$P(v6|v0 = 1, v1 = 0) = [0.04, 0.96]$$

Next, we need to find the marginal distribution of $v0$ given the intervention $\text{do}(v1 = 0)$. The intervention does not change the distribution of $v0$ directly, so we use the original CPT for $v0$.
The CPT for $v0$ is given by:

$$P(v0|v1 = 0, v5 = 0) = [0.61, 0.39]$$

$$P(v0|v1 = 0, v5 = 1) = [0.47, 0.53]$$

We need to consider the distribution of $v5$:

$$P(v5) = [0.24, 0.76]$$

Given $v1 = 0$, we need to find the marginal distribution of $v0$:

$$P(v0 = 0) = P(v0 = 0|v1 = 0, v5 = 0)P(v5 = 0) + P(v0 = 0|v1 = 0, v5 = 1)P(v5 = 1)$$

$$P(v0 = 0) = (0.61 \times 0.24) + (0.47 \times 0.76)$$

$$P(v0 = 0) = 0.1464 + 0.3572 = 0.5036$$

$$P(v0 = 1) = 1 - P(v0 = 0) = 1 - 0.5036 = 0.4964$$

Now, we can find the marginal distribution of $v6$:

$$P(v6 = 0) = P(v6 = 0|v0 = 0)P(v0 = 0) + P(v6 = 0|v0 = 1)P(v0 = 1)$$

$$P(v6 = 0) = (0.65 \times 0.5036) + (0.04 \times 0.4964)$$

$$P(v6 = 0) = 0.32734 + 0.019856 = 0.347196$$

$$P(v6 = 1) = 1 - P(v6 = 0) = 1 - 0.347196 = 0.652804$$

ANSWER
$[0.3472, 0.6528]$

Figure 14: Example trace with incremental marginalization. This example is from system `rl 32b init`, on a medium size intervention level query. Trace marginalizes first over v5, then over v0.

**Brute Force Marginalization**

THOUGHT PROCESS
To find the marginal distribution of $v1$ given $v9 = 0$, we need to compute $P(v1|v9 = 0)$. This can be done using the law of total probability and the given conditional probability tables (CPTs).
1. **Identify the relevant CPTs and variables:**
- $P(v1|v7, v9)$
- $P(v7|v6)$
- $P(v9|v0, v6)$
- $P(v6|v0)$
- $P(v0)$
2. **Express $P(v1|v9 = 0)$ using the law of total probability:**

$$P(v1|v9 = 0) = \sum_{v7, v6, v0} P(v1|v7, v9 = 0) \cdot P(v7|v6) \cdot P(v9 = 0|v0, v6) \cdot P(v6|v0) \cdot P(v0)$$

3. **Break down the computation:**
- Compute $P(v9 = 0|v0, v6)$ from the CPTs.
- Compute $P(v6|v0)$ from the CPTs.
- Compute $P(v0)$ from the CPTs.
- Compute $P(v7|v6)$ from the CPTs.
- Compute $P(v1|v7, v9 = 0)$ from the CPTs.
4. **Compute each term:**
- $P(v9 = 0|v0, v6)$:

$$P(v9 = 0|v0 = 0, v6 = 0) = 0.95, P(v9 = 0|v0 = 0, v6 = 1) = 0.6,$$

$$P(v9 = 0|v0 = 1, v6 = 0) = 0.31, P(v9 = 0|v0 = 1, v6 = 1) = 0.32$$

- $P(v6|v0)$:

$$P(v6 = 0|v0 = 0) = 0.32, \quad P(v6 = 1|v0 = 0) = 0.68, \quad P(v6 = 0|v0 = 1) = 0.03, \quad P(v6 = 1|v0 = 1) = 0.97$$

- $P(v0)$:

$$P(v0 = 0) = 0.65, \quad P(v0 = 1) = 0.35$$

- $P(v7|v6)$:

$$P(v7 = 0|v6 = 0) = 0.12, \quad P(v7 = 1|v6 = 0) = 0.88, \quad P(v7 = 0|v6 = 1) = 0.71, \quad P(v7 = 1|v6 = 1) = 0.29$$

- $P(v1|v7, v9 = 0)$:

$$P(v1 = 0|v7 = 0, v9 = 0) = 0.2, \quad P(v1 = 1|v7 = 0, v9 = 0) = 0.8,$$

$$P(v1 = 0|v7 = 1, v9 = 0) = 0.03, \quad P(v1 = 1|v7 = 1, v9 = 0) = 0.97$$

5. **Combine all terms:**

$$P(v1 = 0|v9 = 0) = \sum_{v7, v6, v0} P(v1 = 0|v7, v9 = 0) \cdot P(v7|v6) \cdot P(v9 = 0|v0, v6) \cdot P(v6|v0) \cdot P(v0)$$

$$P(v1 = 1|v9 = 0) = \sum_{v7, v6, v0} P(v1 = 1|v7, v9 = 0) \cdot P(v7|v6) \cdot P(v9 = 0|v0, v6) \cdot P(v6|v0) \cdot P(v0)$$

6. **Calculate the sums:**

$$P(v1 = 0|v9 = 0) = (0.2 \cdot 0.12 \cdot 0.95 \cdot 0.32 \cdot 0.65 + 0.2 \cdot 0.88 \cdot 0.6 \cdot 0.68 \cdot 0.65 + 0.03 \cdot 0.12 \cdot 0.31 \cdot 0.03 \cdot 0.35 + 0.03 \cdot 0.88 \cdot 0.32 \cdot 0.97 \cdot 0.35)$$

$$P(v1 = 1|v9 = 0) = (0.8 \cdot 0.12 \cdot 0.95 \cdot 0.32 \cdot 0.65 + 0.8 \cdot 0.88 \cdot 0.6 \cdot 0.68 \cdot 0.65 + 0.97 \cdot 0.12 \cdot 0.31 \cdot 0.03 \cdot 0.35 + 0.97 \cdot 0.88 \cdot 0.32 \cdot 0.97 \cdot 0.35)$$

7. **Perform the calculations:**

$$P(v1 = 0|v9 = 0) = 0.002496 + 0.030912 + 0.0000324 + 0.0078528 = 0.0412932$$

$$P(v1 = 1|v9 = 0) = 0.019968 + 0.35808 + 0.0003816 + 0.285632 = 0.6640622$$

8. **Round to two decimal places:**

$$P(v1 = 0|v9 = 0) \approx 0.04, \quad P(v1 = 1|v9 = 0) \approx 0.66$$

ANSWER
$[0.04, 0.66]$

Figure 15: Example Trace with brute force marginalization. This example is from system `rl 7b init`, on a medium size association level query.

**Neighbor Marginalization**

THOUGHT PROCESS
1. **Intervention on $v2$:** We need to compute the marginal distribution of $v1$ after setting $v2 = 0$. This means we need to consider the distribution of $v1$ given $v2 = 0$ and sum over all possible values of $v5$ (since $v5$ is a parent of $v1$).
2. **CPT for $v1$ given $v2 = 0$:** The CPT for $v1$ given $v2$ and $v5$ is:
- $P(v1|v2 = 0, v5 = 0) = [0.83, 0.17]$
- $P(v1|v2 = 0, v5 = 1) = [0.22, 0.78]$
3. **Marginalize over $v5$:** We need to sum over the probabilities of $v5$ (which is $P(v5) = [0.49, 0.51]$).
4. **Compute the marginal distribution:**
- $P(v1 = 0|v2 = 0) = P(v1 = 0|v2 = 0, v5 = 0) \cdot P(v5 = 0) + P(v1 = 0|v2 = 0, v5 = 1) \cdot P(v5 = 1)$
- $P(v1 = 1|v2 = 0) = P(v1 = 1|v2 = 0, v5 = 0) \cdot P(v5 = 0) + P(v1 = 1|v2 = 0, v5 = 1) \cdot P(v5 = 1)$
Let's compute these values step by step.

$$P(v1 = 0|v2 = 0) = 0.83 \cdot 0.49 + 0.22 \cdot 0.51 = 0.4067 + 0.1112 = 0.5179$$

$$P(v1 = 1|v2 = 0) = 0.17 \cdot 0.49 + 0.78 \cdot 0.51 = 0.0833 + 0.3978 = 0.4811$$

Rounding to two decimal places, we get:

$$P(v1 = 0|v2 = 0) = 0.52$$

$$P(v1 = 1|v2 = 0) = 0.48$$

ANSWER
$[0.52, 0.48]$

Figure 16: Example trace with summation over immediate parents. This example is from system `rl 7b init`, on a medium size intervention level query. These solutions introduce ambiguity between brute-force summation and incremental summation since they are the trivial cases of both. Thus, we treat them as a separate category.

**No Marginalization**

THOUGHT PROCESS
The question asks for the marginal distribution of v7 given that v4=1. This means we need to compute the probability distribution of v7 without conditioning on v4, but we can use the conditional probability of v4 given v5 to find the marginal distribution of v7.
ANSWER
$[0.31, 0.69]$

Figure 17: Example trace with no explicit marginalization. This example is from system `rl 3b`, on a small size association level query. Trace is inconsistent, and contains an error about ignoring v4.

**3B Init's Reasoning Traces are Error Prone**

THOUGHT PROCESS

To find the marginal distribution of v2 given that v8 is set to 1, we need to compute the probability distribution of v2 over its possible values (0 and 1) under the intervention. This involves marginalizing out the effect of v8 from the joint distribution of v2 and v8.

The formula for the marginal distribution of v2 given v8=1 is:

$$P(v2 = 0|v8 = 1) = \sum_{v4} P(v2 = 0, v8 = 1|v4)P(v4)$$

$$P(v2 = 1|v8 = 1) = \sum_{v4} P(v2 = 1, v8 = 1|v4)P(v4)$$

We need to compute the conditional probabilities $P(v2 = 0, v8 = 1|v4)$ and $P(v2 = 1, v8 = 1|v4)$ using the CPTs for v2 and v8.

Let's start with $P(v2 = 0, v8 = 1|v4)$: treated intervention as observation

$$P(v2 = 0, v8 = 1|v4) = P(v2 = 0|v8 = 1, v4)P(v8 = 1|v4)$$
$$P(v2 = 1, v8 = 1|v4) = P(v2 = 1|v8 = 1, v4)P(v8 = 1|v4)$$

From the CPTs:

$$P(v2 = 0|v8 = 1, v4) = 0.97$$
$$P(v2 = 1|v8 = 1, v4) = 0.03$$
$$P(v8 = 1|v4) = 0.64$$

Thus: assumed v4 = 1 here and didn't do v4 = 0

$$P(v2 = 0, v8 = 1|v4) = 0.97 \times 0.64 = 0.6208$$

$$P(v2 = 1, v8 = 1|v4) = 0.03 \times 0.64 = 0.0192$$

Now, we need to marginalize these over v4:

$$P(v2 = 0|v8 = 1) = \sum_{v4} P(v2 = 0, v8 = 1|v4)P(v4)$$

$$P(v2 = 1|v8 = 1) = \sum_{v4} P(v2 = 1, v8 = 1|v4)P(v4)$$

Given the CPT for v4:

$$P(v4 = 0) = 0.51$$
$$P(v4 = 1) = 0.49$$

So: invalid marginalization formula

$$P(v2 = 0|v8 = 1) = 0.6208 \times 0.51 + 0.0192 \times 0.49 = 0.316608 + 0.009408 = 0.326016$$
$$P(v2 = 1|v8 = 1) = 0.0192 \times 0.51 + 0.6208 \times 0.49 = 0.009792 + 0.299192 = 0.308984$$

arithmetic error with $0.6208 \times 0.49$
Rounding to the nearest two digits:

$$P(v2 = 0|v8 = 1) \approx 0.33$$
$$P(v2 = 1|v8 = 1) \approx 0.31$$

ANSWER

$$0.33, 0.31$$

Figure 18: Example trace illustrating probability errors and arithmetic errors made by 3b model *before* RLVR for intervention query $p(v_{2_{(v_8=1)}})$. Errors are annotated in red.

# D    ADDITIONAL RESULTS

## D.1    ACCURACY RESULTS

**Accuracy Results** In table 1 we show within-level generalization on filtered test set. In table 2 we show within-level generalization on unfiltered test set. In table 3 we show results for across-level generalization for filtered set. In table 4 we show results for across-level generalization for unfiltered set.

| | interv n10v2 easy (n=1086) | interv n10v2 medium (n=664) | interv n10v2 hard (n=348) | interv n10v2 all (n=2098) | assoc n10v2 easy (n=1684) | assoc n10v2 medium (n=1868) | assoc n10v2 hard (n=388) | assoc n10v2 all (n=3940) | counte n10v2 easy (n=24) | counte n10v2 medium (n=457) | counte n10v2 hard (n=231) | counte n10v2 all (n=712) |
|---|---|---|---|---|---|---|---|---|---|---|---|---|
| rl 32b | **100.000** | **99.398** | **85.632** | **97.426** | **96.793** | **68.201** | **43.299** | **77.970** | 4.167 | **17.068** | **16.450** | **16.433** |
| rl 32b cur | **100.000** | **99.096** | **84.195** | **97.092** | **95.843** | 65.899 | **39.433** | 76.091 | 0.000 | **16.849** | **16.883** | **16.292** |
| sft 32b | 99.724 | 45.934 | 22.126 | 69.828 | 74.347 | 29.336 | 23.454 | 47.995 | **29.167** | **19.037** | 15.152 | **18.118** |
| rl 7b | **99.908** | **91.566** | **58.621** | **90.419** | **81.354** | **40.685** | **33.505** | **57.360** | 4.167 | 13.348 | 15.152 | 13.624 |
| rl 7b cur | **99.724** | **91.566** | 54.023 | **89.561** | **80.523** | **40.096** | **35.309** | **56.904** | 0.000 | 14.004 | 14.719 | 13.764 |
| sft 7b | 99.079 | 13.102 | 16.379 | 58.151 | 67.221 | 21.306 | 15.979 | 40.406 | **20.833** | **14.880** | **15.584** | **15.309** |
| rl 3b | 89.779 | **8.584** | 5.172 | 50.048 | 47.268 | 10.278 | 12.887 | 26.345 | **4.167** | 7.221 | 6.926 | 7.022 |
| rl 3b cur | **97.145** | 8.133 | 5.172 | 53.718 | **62.589** | 10.011 | 10.309 | 32.513 | 0.000 | 7.440 | 6.926 | 7.022 |
| sft 3b | **97.698** | **12.349** | **11.782** | **56.435** | **63.420** | **15.150** | **17.268** | **35.990** | 0.000 | 10.284 | 12.554 | 10.674 |

Table 1: Within level generalization (test, filtered). System accuracy (average CORRECT, see eq. (3)) when training and evaluating on queries from same level. Stratified by query level, and difficulty within each level, as measured by $|V_{\text{rel}}|$, the *size of the relevant subgraph* to the query variable. Note that difficulty is not comparable across different levels. The models are trained on a mix of small/medium/large questions. Systems not significantly worse than the best (with a monte-carlo paired permutation test with n=10000) are bolded.

| | interv n10v2 easy (n=4661) | interv n10v2 medium (n=2428) | interv n10v2 hard (n=911) | interv n10v2 all (n=8000) | assoc n10v2 easy (n=3398) | assoc n10v2 medium (n=3831) | assoc n10v2 hard (n=771) | assoc n10v2 all (n=8000) | counte n10v2 easy (n=2440) | counte n10v2 medium (n=4533) | counte n10v2 hard (n=1027) | counte n10v2 all (n=8000) |
|---|---|---|---|---|---|---|---|---|---|---|---|---|
| rl 32b | **99.850** | **95.140** | **76.729** | **95.788** | **98.411** | **79.953** | **53.567** | **85.250** | **93.975** | **69.446** | **41.383** | **73.325** |
| rl 32b cur | 99.592 | 93.781 | **76.400** | 95.188 | 97.793 | 77.839 | 48.768 | 83.513 | **93.525** | **69.667** | **42.454** | **73.450** |
| sft 32b | 93.177 | 37.068 | 26.894 | 68.600 | 84.255 | 47.612 | 36.187 | 62.075 | 86.926 | 50.849 | 29.893 | 59.162 |
| rl 7b | **99.657** | **85.502** | **54.226** | **90.188** | **90.318** | **64.213** | **46.304** | **73.575** | **87.295** | **61.196** | **38.267** | **66.212** |
| rl 7b cur | **99.378** | **83.979** | **51.811** | **89.287** | **89.670** | **64.161** | **48.898** | **73.525** | **86.762** | **62.453** | **38.462** | **66.787** |
| sft 7b | 85.518 | 18.328 | 22.722 | 57.975 | 77.340 | 37.745 | 26.070 | 53.438 | 80.492 | 41.562 | 22.590 | 51.000 |
| rl 3b | 76.958 | 7.661 | 6.806 | 47.938 | 63.420 | 22.971 | 17.510 | 39.625 | 66.025 | 29.385 | 11.490 | 38.263 |
| rl 3b cur | 78.502 | 7.372 | 6.806 | 48.750 | 71.307 | 22.240 | 14.656 | 42.350 | 74.877 | 28.877 | 11.977 | 40.737 |
| sft 3b | **79.682** | **11.656** | **14.929** | **51.663** | **72.484** | **29.313** | **23.217** | **47.062** | **77.664** | **32.274** | **18.306** | **44.325** |

Table 2: Within level generalization (test, unfiltered). System accuracy (average CORRECT, see eq. (3)) when training and evaluating on queries from same level. Stratified by query level, and difficulty within each level, as measured by $|V_{\text{rel}}|$, the *size of the relevant subgraph* to the query variable. Note that difficulty is not comparable across different levels. The models are trained on a mix of small/medium/large questions. Systems not significantly worse than the best (with a monte-carlo paired permutation test with n=10000) are bolded.

| | interv n10v2 easy (n=1086) | interv n10v2 medium (n=664) | interv n10v2 hard (n=348) | interv n10v2 all (n=2098) | | assoc n10v2 easy (n=1684) | assoc n10v2 medium (n=1868) | assoc n10v2 hard (n=388) | assoc n10v2 all (n=3940) |
|---|---|---|---|---|---|---|---|---|---|
| rl 32b asso | **100.000** | **99.398** | **85.920** | **97.474** | rl 32b inte | **95.487** | **62.152** | **38.660** | **74.086** |
| rl 32b coun | **100.000** | 98.946 | 80.172 | 96.378 | rl 32b coun | **96.378** | **62.152** | 34.794 | **74.086** |
| 32b rl init | **99.908** | 91.867 | 52.011 | 89.418 | 32b rl init | 88.539 | 39.186 | 22.680 | 58.655 |
| sft 32b asso | **99.724** | 41.717 | 21.552 | 68.398 | sft 32b inte | 72.031 | 26.927 | 24.485 | 45.964 |
| sft 32b coun | **99.908** | 44.428 | 26.437 | 70.162 | sft 32b coun | 57.423 | 26.927 | 22.938 | 39.569 |
| 32b sft init | 95.580 | 12.199 | 12.356 | 55.386 | 32b sft init | 58.314 | 11.991 | 10.825 | 31.675 |

| | | | | counte n10v2 easy (n=24) | counte n10v2 medium (n=457) | counte n10v2 hard (n=231) | counte n10v2 all (n=712) |
|---|---|---|---|---|---|---|---|
| | | | rl 32b inte | **0.000** | **17.287** | **16.883** | **16.573** |
| | | | rl 32b asso | **4.167** | **17.943** | **19.913** | **18.118** |
| | | | 32b rl init | **0.000** | 12.473 | 9.091 | 10.955 |
| | | | sft 32b inte | **8.333** | **14.004** | **13.420** | **13.624** |
| | | | sft 32b asso | **4.167** | 11.160 | 12.554 | 11.376 |
| | | | 32b sft init | **4.167** | 8.315 | 7.359 | 7.865 |

| | interv n10v2 easy (n=1086) | interv n10v2 medium (n=664) | interv n10v2 hard (n=348) | interv n10v2 all (n=2098) | | assoc n10v2 easy (n=1684) | assoc n10v2 medium (n=1868) | assoc n10v2 hard (n=388) | assoc n10v2 all (n=3940) |
|---|---|---|---|---|---|---|---|---|---|
| rl 7b asso | **98.619** | **87.500** | **52.874** | **87.512** | rl 7b inte | 48.337 | **34.904** | **25.258** | **39.695** |
| rl 7b coun | **99.079** | **87.199** | 44.540 | **86.273** | rl 7b coun | **55.641** | 31.156 | **23.454** | **40.863** |
| 7b rl init | 61.510 | 51.958 | 16.092 | 50.953 | 7b rl init | 29.513 | 11.135 | 6.701 | 18.553 |
| sft 7b asso | **99.263** | 19.729 | 18.391 | 60.677 | sft 7b inte | **56.116** | 18.094 | **21.907** | 34.721 |
| sft 7b coun | 97.514 | 18.825 | 22.126 | 60.105 | sft 7b coun | 31.116 | 18.737 | 18.299 | 23.985 |
| 7b sft init | 24.862 | 4.217 | 0.287 | 14.252 | 7b sft init | 7.898 | 2.677 | 1.546 | 4.797 |

| | | | | counte n10v2 easy (n=24) | counte n10v2 medium (n=457) | counte n10v2 hard (n=231) | counte n10v2 all (n=712) |
|---|---|---|---|---|---|---|---|
| | | | rl 7b inte | **0.000** | **13.567** | **11.688** | 12.500 |
| | | | rl 7b asso | **4.167** | **14.880** | **15.584** | **14.747** |
| | | | 7b rl init | **4.167** | 8.753 | 4.762 | 7.303 |
| | | | sft 7b inte | **8.333** | 6.127 | 6.494 | 6.320 |
| | | | sft 7b asso | **8.333** | 9.409 | **9.957** | 9.551 |
| | | | 7b sft init | **4.167** | 1.751 | 0.433 | 1.404 |

| | interv n10v2 easy (n=1086) | interv n10v2 medium (n=664) | interv n10v2 hard (n=348) | interv n10v2 all (n=2098) | | assoc n10v2 easy (n=1684) | assoc n10v2 medium (n=1868) | assoc n10v2 hard (n=388) | assoc n10v2 all (n=3940) |
|---|---|---|---|---|---|---|---|---|---|
| rl 3b asso | 77.072 | **9.337** | 5.747 | 43.804 | rl 3b inte | **41.627** | 11.456 | 11.340 | 24.340 |
| rl 3b coun | 74.125 | 8.434 | 5.172 | 41.897 | rl 3b coun | 35.926 | 9.636 | 9.794 | 20.888 |
| 3b rl init | 24.309 | **10.392** | 3.736 | 16.492 | 3b rl init | 9.561 | 5.621 | 4.897 | 7.234 |
| sft 3b asso | 60.866 | **10.994** | **12.931** | 37.131 | sft 3b inte | 38.717 | **15.953** | **15.979** | **25.685** |
| sft 3b coun | **93.370** | **11.446** | **14.080** | **54.290** | sft 3b coun | 16.983 | **14.507** | **15.979** | 15.711 |
| 3b sft init | 22.744 | 8.283 | 2.874 | 14.871 | 3b sft init | 14.014 | 8.458 | 10.567 | 11.041 |

| | | | | counte n10v2 easy (n=24) | counte n10v2 medium (n=457) | counte n10v2 hard (n=231) | counte n10v2 all (n=712) |
|---|---|---|---|---|---|---|---|
| | | | rl 3b inte | **0.000** | 5.689 | **8.225** | 6.320 |
| | | | rl 3b asso | **4.167** | 6.565 | **7.359** | 6.742 |
| | | | 3b rl init | **0.000** | 4.595 | 2.165 | 3.652 |
| | | | sft 3b inte | **0.000** | 10.284 | **11.688** | **10.393** |
| | | | sft 3b asso | **4.167** | 9.409 | 10.390 | 9.551 |
| | | | 3b sft init | **0.000** | 4.814 | 3.030 | 4.073 |

Table 3: Across-level generalization (test, filtered). Row specify which level trained on, column specify which level evaluated on. System accuracy (average CORRECT, see eq. (3)) on evaluation sets of different difficulties, as measured by $|V_{\text{rel}}|$, the *size of the relevant subgraph* to the query variable. Note that difficulty is not comparable across different levels. The models are trained on a mix of easy/medium/hard questions. Systems not significantly worse than the best (with a monte-carlo paired permutation test with n=10000) are bolded.

|  | interv n10v2 easy (n=4661) | interv n10v2 medium (n=2428) | interv n10v2 hard (n=911) | interv n10v2 all (n=8000) |  | assoc n10v2 easy (n=3398) | assoc n10v2 medium (n=3831) | assoc n10v2 hard (n=771) | assoc n10v2 all (n=8000) |
|---|---|---|---|---|---|---|---|---|---|
| rl 32b asso | **98.048** | **93.287** | **77.827** | **94.300** | rl 32b inte | **97.705** | **74.915** | **46.433** | **81.850** |
| rl 32b coun | 97.147 | 91.269 | 72.338 | 92.537 | rl 32b coun | **98.175** | **73.532** | 41.634 | 80.925 |
| 32b rl init | 95.559 | 77.636 | 44.566 | 84.312 | 32b rl init | 93.320 | 51.031 | 24.903 | 66.475 |
| sft 32b asso | 87.170 | 36.656 | 27.442 | 65.038 | sft 32b inte | 83.196 | 45.784 | 35.279 | 60.662 |
| sft 32b coun | 83.201 | 39.621 | 31.065 | 64.038 | sft 32b coun | 75.544 | 46.150 | 34.112 | 57.475 |
| 32b sft init | 76.636 | 11.038 | 14.490 | 49.650 | 32b sft init | 67.039 | 23.727 | 16.083 | 41.388 |

|  | counte n10v2 easy (n=2440) | counte n10v2 medium (n=4533) | counte n10v2 hard (n=1027) | counte n10v2 all (n=8000) |
|---|---|---|---|---|
| rl 32b inte | 92.131 | 68.564 | **41.967** | 72.338 |
| rl 32b asso | **93.033** | **70.042** | **43.720** | **73.675** |
| 32b rl init | 88.074 | 55.306 | 28.140 | 61.812 |
| sft 32b inte | 79.426 | 48.577 | 28.627 | 55.425 |
| sft 32b asso | 76.967 | 48.687 | 29.309 | 54.825 |
| 32b sft init | 64.590 | 31.370 | 14.411 | 39.325 |

|  | interv n10v2 easy (n=4661) | interv n10v2 medium (n=2428) | interv n10v2 hard (n=911) | interv n10v2 all (n=8000) |  | assoc n10v2 easy (n=3398) | assoc n10v2 medium (n=3831) | assoc n10v2 hard (n=771) | assoc n10v2 all (n=8000) |
|---|---|---|---|---|---|---|---|---|---|
| rl 7b asso | **98.777** | **79.572** | **46.652** | **87.013** | rl 7b inte | 72.543 | **54.346** | **33.982** | **60.113** |
| rl 7b coun | 98.219 | 76.895 | 39.627 | 85.075 | rl 7b coun | **75.044** | 49.987 | **30.999** | 58.800 |
| 7b rl init | 62.969 | 32.867 | 14.380 | 48.300 | 7b rl init | 37.905 | 14.748 | 7.134 | 23.850 |
| sft 7b asso | 79.897 | 22.858 | 21.625 | 55.950 | sft 7b inte | 70.983 | 34.299 | 29.053 | 49.375 |
| sft 7b coun | 83.802 | 20.099 | 26.015 | 57.888 | sft 7b coun | 58.711 | 35.787 | 29.053 | 44.875 |
| 7b sft init | 18.537 | 1.689 | 0.110 | 11.325 | 7b sft init | 12.919 | 3.211 | 1.686 | 7.187 |

|  | counte n10v2 easy (n=2440) | counte n10v2 medium (n=4533) | counte n10v2 hard (n=1027) | counte n10v2 all (n=8000) |
|---|---|---|---|---|
| rl 7b inte | **84.672** | **62.850** | **34.761** | **65.900** |
| rl 7b asso | **83.934** | 60.379 | **34.664** | 64.263 |
| 7b rl init | 60.615 | 31.568 | 12.561 | 37.988 |
| sft 7b inte | 73.361 | 35.892 | 19.182 | 45.175 |
| sft 7b asso | 65.533 | 40.393 | 19.669 | 45.400 |
| 7b sft init | 31.148 | 9.618 | 1.753 | 15.175 |

|  | interv n10v2 easy (n=4661) | interv n10v2 medium (n=2428) | interv n10v2 hard (n=911) | interv n10v2 all (n=8000) |  | assoc n10v2 easy (n=3398) | assoc n10v2 medium (n=3831) | assoc n10v2 hard (n=771) | assoc n10v2 all (n=8000) |
|---|---|---|---|---|---|---|---|---|---|
| rl 3b asso | 73.868 | 7.784 | 6.476 | 46.137 | rl 3b inte | **60.153** | 23.388 | 17.121 | 38.400 |
| rl 3b coun | 73.203 | 6.755 | 6.257 | 45.413 | rl 3b coun | 57.004 | 21.874 | 15.045 | 36.138 |
| 3b rl init | 35.250 | 6.260 | 4.281 | 22.925 | 3b rl init | 26.574 | 10.232 | 7.393 | 16.900 |
| sft 3b asso | 71.680 | **12.891** | **17.124** | 47.625 | sft 3b inte | **60.212** | **28.974** | **24.125** | **41.775** |
| sft 3b coun | **79.446** | **12.191** | 13.941 | **51.575** | sft 3b coun | 49.205 | 27.982 | 21.401 | 36.362 |
| 3b sft init | 42.502 | 5.890 | 5.488 | 27.175 | 3b sft init | 35.227 | 16.967 | 12.192 | 24.262 |

|  | counte n10v2 easy (n=2440) | counte n10v2 medium (n=4533) | counte n10v2 hard (n=1027) | counte n10v2 all (n=8000) |
|---|---|---|---|---|
| rl 3b inte | **62.459** | 29.164 | 12.658 | 37.200 |
| rl 3b asso | 60.615 | 29.274 | 12.074 | 36.625 |
| 3b rl init | 28.443 | 15.420 | 6.134 | 18.200 |
| sft 3b inte | **62.459** | 31.348 | **15.774** | **38.838** |
| sft 3b asso | 59.713 | **32.120** | **16.456** | **38.525** |
| 3b sft init | 20.902 | 10.170 | 3.603 | 12.600 |

Table 4: Across-level generalization (test, unfiltered). Row specify which level trained on, column specify which level evaluated on. System accuracy (average CORRECT, see eq. (3)) on evaluation sets of different difficulties, as measured by $|V_{\text{rel}}|$, the *size of the relevant subgraph* to the query variable. Note that difficulty is not comparable across different levels. The models are trained on a mix of easy/medium/hard questions. Systems not significantly worse than the best (with a monte-carlo paired permutation test with n=10000) are bolded.

## D.2 HINT RESULTS

**Hint Results** In fig. 19 we show the performance of LLMs after RLVR with hint in system prompt (fig. 9) for the counterfactual level. It did not improve over the simple system prompt (fig. 8).

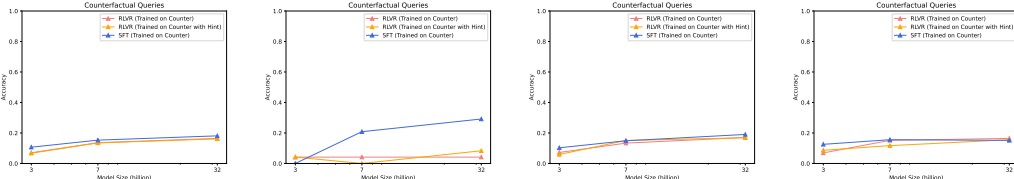

Figure 19: Counterfactual level with hint. Prompting with a hint about how to solve counterfactual queries by twin-network-graph is not enough to induce genuine solutions and improve performance post-RLVR. From left to right is accuracy breakdown on **all**, **small**, **medium**, and finally **large** problems. Having hint in the prompt did not significantly improve RLVR's performance on counterfactual level.

## D.3 LLM JUDGE RESULTS

**LLM Judge Results** In table 5 we show numerical results for strategy categorization, visualized in fig. 5. In table 6 we show numerical results for strategy categorization on unfiltered test set, visualized in fig. 20. In table 7 we show numerical results for strategy categorization on the hard subset of the filtered test set, visualized in fig. 21.

| | reasoning incremental (n=80) | reasoning brute (n=80) | reasoning immediate (n=80) | reasoning none (n=80) | | derivation error yes (n=80) | derivation error no (n=80) | derivation error na (n=80) |
|---|---|---|---|---|---|---|---|---|
| rl 3b | 0.000 | 0.000 | 20.000 | 80.000 | rl 3b | 41.250 | 31.250 | 27.500 |
| rl 3b init | 8.750 | 3.750 | 51.250 | 36.250 | rl 3b init | 80.000 | 17.500 | 2.500 |
| rl 7b | 32.500 | 0.000 | 35.000 | 32.500 | rl 7b | 16.250 | 83.750 | 0.000 |
| rl 7b init | 25.000 | 1.250 | 46.250 | 27.500 | rl 7b init | 40.000 | 57.500 | 2.500 |
| rl 32b | 33.750 | 0.000 | 36.250 | 30.000 | rl 32b | 5.000 | 95.000 | 0.000 |
| rl 32b init | 28.750 | 5.000 | 33.750 | 32.500 | rl 32b init | 5.000 | 95.000 | 0.000 |
| | reasoning incremental (n=80) | reasoning brute (n=80) | reasoning immediate (n=80) | reasoning none (n=80) | | derivation error yes (n=80) | derivation error no (n=80) | derivation error na (n=80) |
| rl 3b | 0.000 | 0.000 | 17.500 | 82.500 | rl 3b | 46.250 | 23.750 | 28.750 |
| rl 3b init | 6.250 | 8.750 | 66.250 | 18.750 | rl 3b init | 85.000 | 10.000 | 5.000 |
| rl 7b | 32.500 | 0.000 | 37.500 | 30.000 | rl 7b | 50.000 | 50.000 | 0.000 |
| rl 7b init | 12.500 | 18.750 | 67.500 | 1.250 | rl 7b init | 67.500 | 32.500 | 0.000 |
| rl 32b | 46.250 | 3.750 | 28.750 | 21.250 | rl 32b | 27.500 | 71.250 | 1.250 |
| rl 32b init | 22.500 | 17.500 | 38.750 | 21.250 | rl 32b init | 28.750 | 68.750 | 2.500 |
| | reasoning incremental (n=80) | reasoning brute (n=80) | reasoning immediate (n=80) | reasoning none (n=80) | | derivation error yes (n=80) | derivation error no (n=80) | derivation error na (n=80) |
| rl 3b | 0.000 | 0.000 | 17.500 | 82.500 | rl 3b | 78.750 | 6.250 | 13.750 |
| rl 3b init | 7.500 | 6.250 | 46.250 | 40.000 | rl 3b init | 92.500 | 2.500 | 5.000 |
| rl 7b | 41.250 | 2.500 | 47.500 | 8.750 | rl 7b | 88.750 | 11.250 | 0.000 |
| rl 7b init | 26.250 | 5.000 | 55.000 | 13.750 | rl 7b init | 93.750 | 6.250 | 0.000 |
| rl 32b | 45.000 | 8.750 | 38.750 | 7.500 | rl 32b | 77.500 | 22.500 | 0.000 |
| rl 32b init | 32.500 | 11.250 | 43.750 | 11.250 | rl 32b init | 83.750 | 15.000 | 1.250 |

Table 5: LLM Judge Numerical Results on Filtered Set. See fig. 5 for visualization. Top to bottom: intervention, association, counterfactual.

| | reasoning incremental (n=80) | reasoning brute (n=80) | reasoning immediate (n=80) | reasoning none (n=80) | | derivation error yes (n=80) | derivation error no (n=80) | derivation error na (n=80) |
|---|---|---|---|---|---|---|---|---|
| rl 3b | 0.000 | 0.000 | 12.500 | 87.500 | 50.000 | 28.750 | 20.000 |
| rl 3b init | 10.000 | 5.000 | 51.250 | 33.750 | 92.500 | 5.000 | 2.500 |
| rl 7b | 25.000 | 3.750 | 36.250 | 35.000 | 17.500 | 81.250 | 0.000 |
| rl 7b init | 20.000 | 7.500 | 46.250 | 25.000 | 60.000 | 40.000 | 0.000 |
| rl 32b | 27.500 | 1.250 | 41.250 | 28.750 | 6.250 | 92.500 | 1.250 |
| rl 32b init | 26.250 | 1.250 | 40.000 | 32.500 | 11.250 | 88.750 | 0.000 |

| | reasoning incremental (n=80) | reasoning brute (n=80) | reasoning immediate (n=80) | reasoning none (n=80) | | derivation error yes (n=80) | derivation error no (n=80) | derivation error na (n=80) |
|---|---|---|---|---|---|---|---|---|
| rl 3b | 0.000 | 0.000 | 7.500 | 92.500 | 53.750 | 27.500 | 18.750 |
| rl 3b init | 11.250 | 7.500 | 52.500 | 28.750 | 88.750 | 8.750 | 2.500 |
| rl 7b | 28.750 | 0.000 | 37.500 | 33.750 | 40.000 | 57.500 | 2.500 |
| rl 7b init | 20.000 | 23.750 | 46.250 | 10.000 | 72.500 | 27.500 | 0.000 |
| rl 32b | 36.250 | 2.500 | 31.250 | 30.000 | 16.250 | 81.250 | 1.250 |
| rl 32b init | 30.000 | 11.250 | 27.500 | 31.250 | 16.250 | 80.000 | 3.750 |

| | reasoning incremental (n=80) | reasoning brute (n=80) | reasoning immediate (n=80) | reasoning none (n=80) | | derivation error yes (n=80) | derivation error no (n=80) | derivation error na (n=80) |
|---|---|---|---|---|---|---|---|---|
| rl 3b | 0.000 | 0.000 | 10.000 | 90.000 | 45.000 | 22.500 | 30.000 |
| rl 3b init | 6.250 | 0.000 | 46.250 | 47.500 | 81.250 | 10.000 | 7.500 |
| rl 7b | 18.750 | 0.000 | 40.000 | 41.250 | 46.250 | 52.500 | 1.250 |
| rl 7b init | 17.500 | 6.250 | 36.250 | 40.000 | 70.000 | 26.250 | 2.500 |
| rl 32b | 31.250 | 5.000 | 33.750 | 30.000 | 36.250 | 63.750 | 0.000 |
| rl 32b init | 18.750 | 10.000 | 25.000 | 46.250 | 45.000 | 47.500 | 6.250 |

Table 6: LLM Judge Numerical Results on Unfiltered Set. See fig. 20 for visualization. Top to bottom: intervention, association, counterfactual.

| | reasoning incremental (n=80) | reasoning brute (n=80) | reasoning immediate (n=80) | reasoning none (n=80) | | derivation error yes (n=80) | derivation error no (n=80) | derivation error na (n=80) |
|---|---|---|---|---|---|---|---|---|
| rl 3b | 0.000 | 0.000 | 3.750 | 96.250 | 61.250 | 0.000 | 36.250 |
| rl 3b init | 11.250 | 12.500 | 48.750 | 27.500 | 95.000 | 0.000 | 5.000 |
| rl 7b | 97.500 | 0.000 | 2.500 | 0.000 | 47.500 | 52.500 | 0.000 |
| rl 7b init | 40.000 | 6.250 | 52.500 | 1.250 | 85.000 | 13.750 | 1.250 |
| rl 32b | 93.750 | 5.000 | 1.250 | 0.000 | 32.500 | 67.500 | 0.000 |
| rl 32b init | 82.500 | 15.000 | 1.250 | 1.250 | 28.750 | 70.000 | 1.250 |

| | reasoning incremental (n=80) | reasoning brute (n=80) | reasoning immediate (n=80) | reasoning none (n=80) | | derivation error yes (n=80) | derivation error no (n=80) | derivation error na (n=80) |
|---|---|---|---|---|---|---|---|---|
| rl 3b | 0.000 | 0.000 | 6.250 | 93.750 | 48.750 | 1.250 | 48.750 |
| rl 3b init | 11.250 | 10.000 | 47.500 | 31.250 | 91.250 | 0.000 | 8.750 |
| rl 7b | 67.500 | 1.250 | 20.000 | 11.250 | 90.000 | 10.000 | 0.000 |
| rl 7b init | 11.250 | 62.500 | 23.750 | 2.500 | 91.250 | 8.750 | 0.000 |
| rl 32b | 65.000 | 27.500 | 5.000 | 2.500 | 73.750 | 26.250 | 0.000 |
| rl 32b init | 26.250 | 56.250 | 12.500 | 5.000 | 68.750 | 25.000 | 6.250 |

| | reasoning incremental (n=80) | reasoning brute (n=80) | reasoning immediate (n=80) | reasoning none (n=80) | | derivation error yes (n=80) | derivation error no (n=80) | derivation error na (n=80) |
|---|---|---|---|---|---|---|---|---|
| rl 3b | 0.000 | 0.000 | 8.750 | 91.250 | 66.250 | 1.250 | 30.000 |
| rl 3b init | 13.750 | 2.500 | 28.750 | 55.000 | 85.000 | 2.500 | 12.500 |
| rl 7b | 57.500 | 3.750 | 35.000 | 3.750 | 85.000 | 15.000 | 0.000 |
| rl 7b init | 23.750 | 13.750 | 50.000 | 12.500 | 96.250 | 3.750 | 0.000 |
| rl 32b | 71.250 | 7.500 | 18.750 | 2.500 | 76.250 | 23.750 | 0.000 |
| rl 32b init | 45.000 | 17.500 | 22.500 | 15.000 | 75.000 | 15.000 | 10.000 |

Table 7: LLM Judge Numerical Results on Large Complexity Split of Filtered Test Set. See fig. 21 for visualization. Top to bottom: intervention, association, counterfactual.

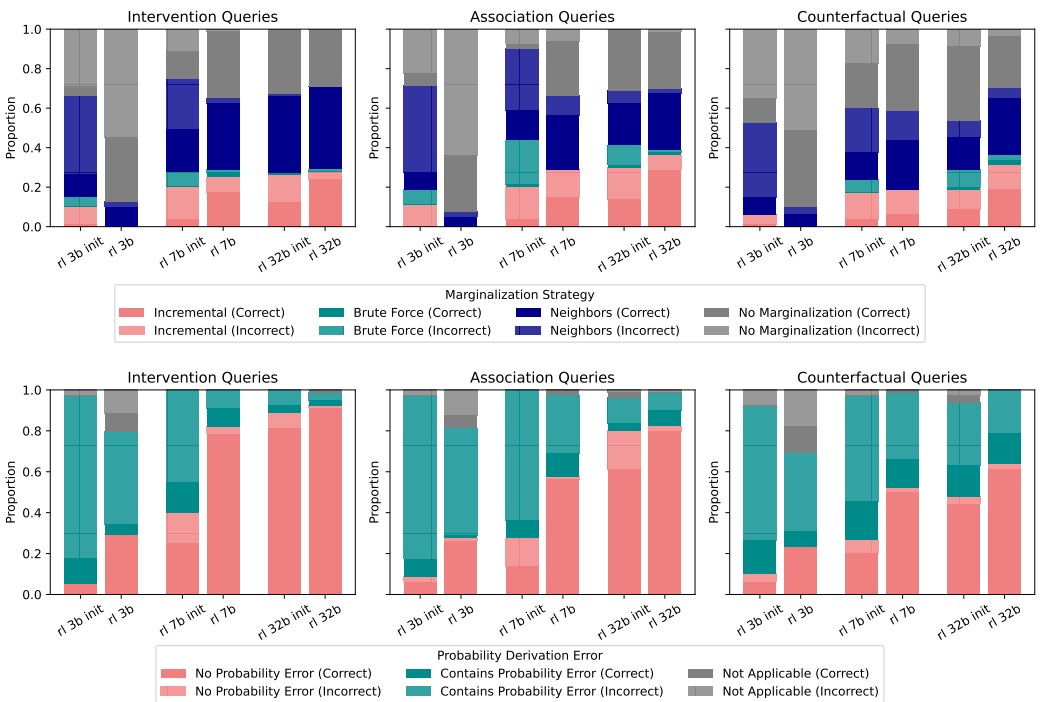

Figure 20: LLM judge analysis of reasoning strategy (top) and existence of derivation errors (bottom) before and after RLVR on **unfiltered** test set. Marginalization strategies are annotated on 80 samples per level. Derivation errors are also annotated on the same 80 samples per level.

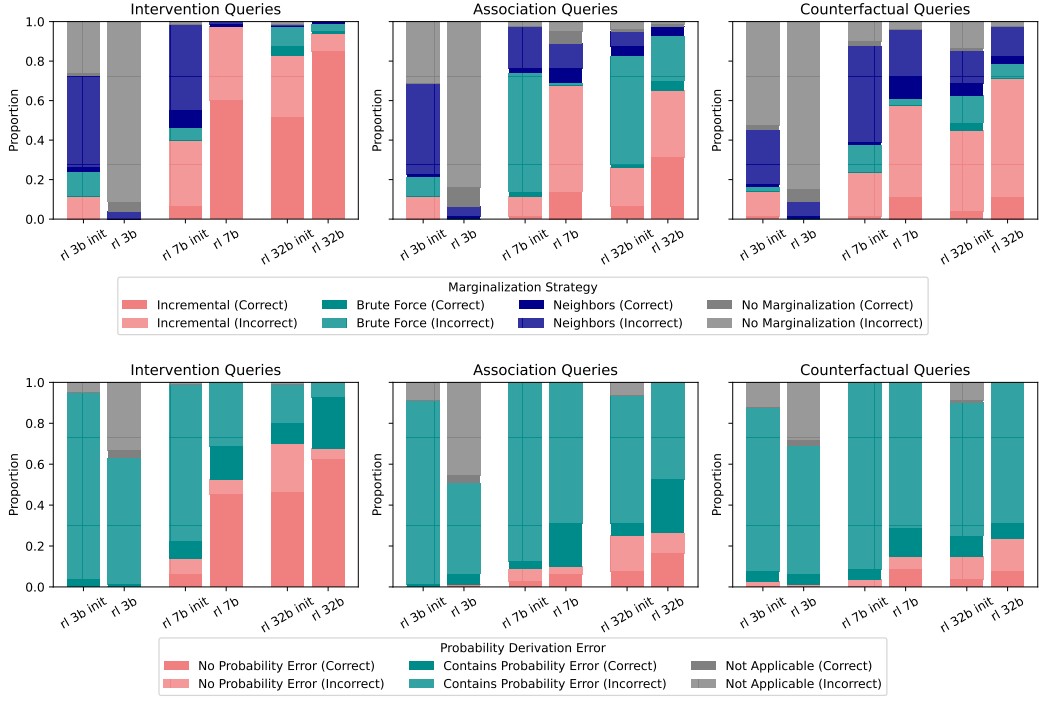

Figure 21: LLM judge analysis of reasoning strategy (top) and existence of derivation errors (bottom) before and after RLVR on the **hard** complexity queries. Marginalization strategies are annotated on 80 samples per level. Derivation errors are also annotated on the same 80 samples per level.

## D.4 PRECISION RESULTS

**Precision Results** We plot precision of SFT vs. RL for all sizes and all levels in figs. 22 to 24.

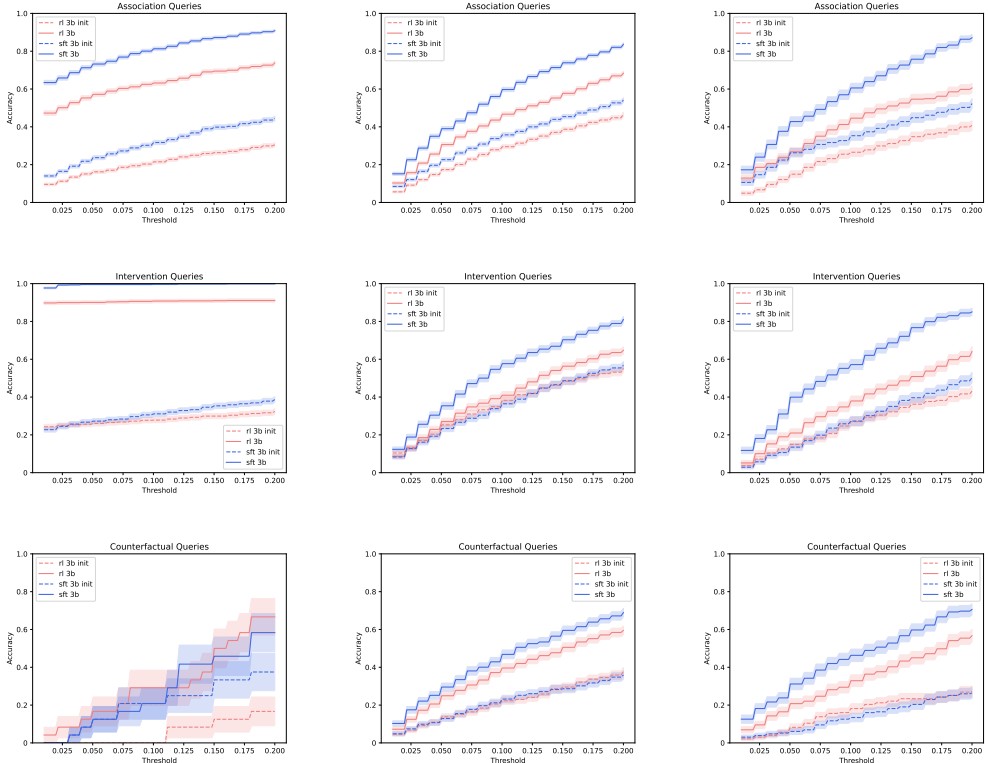

Figure 22: Accuracy by threshold $t \in (0, 0.2]$ for 3B Models. $x$-axis plots threshold for accuracy $t$ (the lower the stricter). $y$-axis is accuracy. From top to bottom are different levels. From left to right are query complexities small, medium, large within each level.

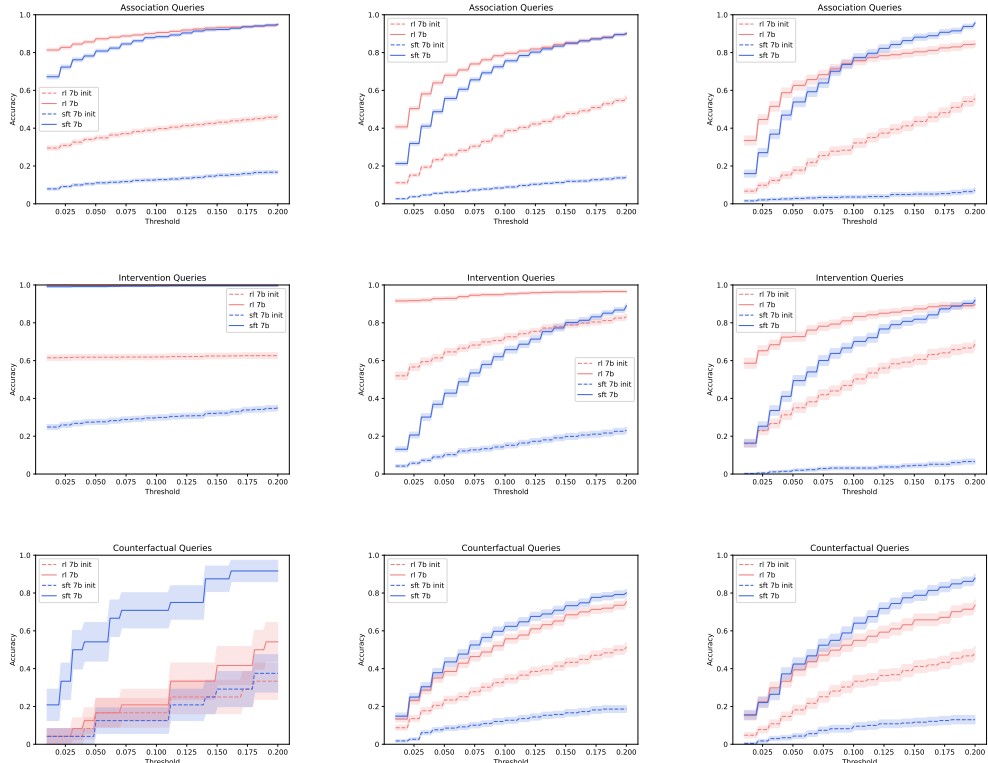

Figure 23: Accuracy by threshold $t \in (0, 0.2]$ for 7B Models. $x$-axis plots threshold for accuracy $t$ (the lower the stricter). $y$-axis is accuracy. From top to bottom are different levels. From left to right are query complexities small, medium, large within each level.

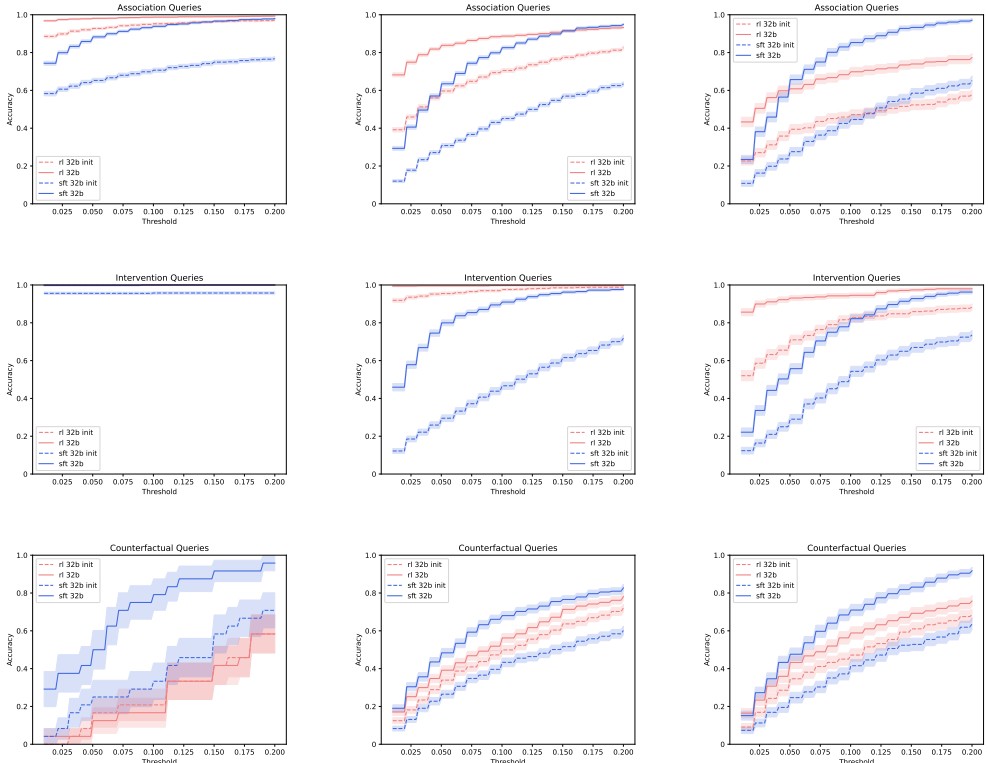

Figure 24: Accuracy by threshold $t \in (0, 0.2]$ for 32B Models. $x$-axis plots threshold for accuracy $t$ (the lower the stricter). $y$-axis is accuracy. From top to bottom are different levels. From left to right are query complexities small, medium, large within each level.

## D.5 SFT WITH REJECTION-SAMPLED COT

**SFT with Rejection-sampled CoT** In our main experiment (section 4.2, fig. 3), we found that 7b and 32b LLMs fine-tuned with RLVR generalizes significantly better than those fine-tuned with SFT on association and intervention level problems. How much of this gap is due to RLVR training on reasoning chains (which our SFT setup did not have)? How much of this gap is due to RLVR training on reasoning chains that are generated on-policy?

To understand the relative contributions of fine-tuning with reasoning chains and fine-tuning with on-policy data, we additionally fine-tuned Qwen2.5-7B-Instruct models on *correct* reasoning chains collected via rejection sampling from Qwen2.5-7B-Instruct itself. We perform rejection sampling across intervention, association, and counterfactual level problmes, and build a supervised fine-tuning dataset for each level with reasoning chains for the same 8000 training problems that are used to perform RLVR and SFT in section 4. For each problem, we independently sample 32 reasoning chains at temperature 1.0 from Qwen2.5-7B-Instruct, and keep only chains-of-thoughts that have a correct final answer. See fig. 25 for a scatter plot of the proportion of correct reasoning chains (out of 32) for a problem versus the difficulty of the problem. We then fine-tune Qwen2.5-7B-Instruct for 5 epochs, and pick the checkpoint that has the highest accuracy on the dev set (described in section 4).

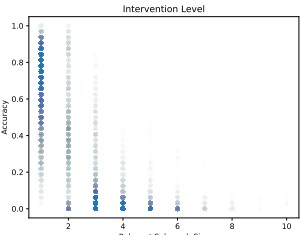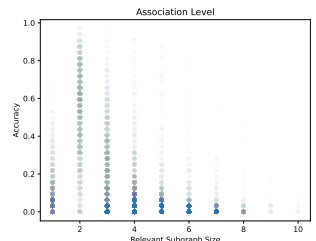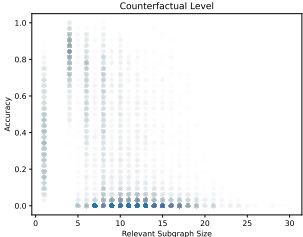

Figure 25: Scatter plot of accuracy by $|V_{\text{rel}}|$, the *size of the relevant subgraph* to the query variable. Each point is a particular query from the (unfiltered) training set.

*Within-level generalization* of SFT on rejection-sampled reasoning chains (RS32) across three levels are included table 8. Comparing RS32 with COT INIT shows that fine-tuning on correct reasoning chains improves the within-level accuracy of the LLM. Comparing RS32 with RL shows that it nonetheless underperforms RLVR, which uses online rather than offline data, and especially so on more difficult splits. Comparing RS32 with direct prediction SFT (DP), we see that fine-tuning with reasoning chains outperforms slightly on association and intervention levels, and is comparable on counterfactual. Overall, these results show that fine-tuning on reasoning-chains and the on-policy nature of RLVR both contribute to its superior performance on within-level generalization.

|  | interv n10v2 easy (n=1086) | interv n10v2 medium (n=664) | interv n10v2 hard (n=348) | interv n10v2 all (n=2098) | assoc n10v2 easy (n=1684) | assoc n10v2 medium (n=1868) | assoc n10v2 hard (n=388) | assoc n10v2 all (n=3940) | counte n10v2 easy (n=24) | counte n10v2 medium (n=457) | counte n10v2 hard (n=231) | counte n10v2 all (n=712) |
|---|---|---|---|---|---|---|---|---|---|---|---|---|
| 7b rl | **99.908** | **91.566** | **58.621** | **90.419** | **81.354** | **40.685** | **33.505** | **57.360** | 4.167 | 13.348 | **15.152** | 13.624 |
| 7b rs32 sft | 96.593 | 70.331 | 21.552 | 75.834 | 69.418 | 19.647 | 14.433 | 40.406 | 0.000 | **10.941** | 9.957 | 10.253 |
| 7b cot init | 61.510 | 51.958 | 16.092 | 50.953 | 29.513 | 11.135 | 6.701 | 18.553 | 4.167 | 8.753 | 4.762 | 7.303 |
| 7b dp sft | 99.079 | 13.102 | 16.379 | 58.151 | 67.221 | 21.306 | 15.979 | 40.406 | **20.833** | **14.880** | **15.584** | **15.309** |
| 7b dp init | 24.862 | 4.217 | 0.287 | 14.252 | 7.898 | 2.677 | 1.546 | 4.797 | 4.167 | 1.751 | 0.433 | 1.404 |

Table 8: Within level generalization (test, filtered) of 7b models fine-tuned with various supervision sources: online chains-of-thought (rl), offline chains-of-thought (rs32), answer only (dp). System accuracy (average CORRECT, see eq. (3)) when training and evaluating on queries from same level. Stratified by query level, and difficulty within each level, as measured by $|V_{\text{rel}}|$, the *size of the relevant subgraph* to the query variable. Note that difficulty is not comparable across different levels. The models are trained on a mix of small/medium/large questions. Systems not significantly worse than the best (with a monte-carlo paired permutation test with n=10000) are bolded. rs32 is fine-tuned on chains-of-thought collected via sampling 32 times per training problem and keeping only correct ones.

| | interv n10v2 easy (n=1086) | interv n10v2 medium (n=664) | interv n10v2 hard (n=348) | interv n10v2 all (n=2098) |
|---|---|---|---|---|
| 7b rl asso | **98.619** | **87.500** | **52.874** | **87.512** |
| 7b sft rs32 asso | 89.963 | 63.404 | 21.264 | 70.162 |
| 7b rl coun | **99.079** | **87.199** | 44.540 | **86.273** |
| 7b sft rs32 coun | 94.383 | 73.343 | 22.701 | 75.834 |
| 7b cot init | 61.510 | 51.958 | 16.092 | 50.953 |
| 7b dp sft asso | **99.263** | 19.729 | 18.391 | 60.677 |
| 7b dp sft coun | 97.514 | 18.825 | 22.126 | 60.105 |
| 7b dp init | 24.862 | 4.217 | 0.287 | 14.252 |
| | assoc n10v2 easy (n=1684) | assoc n10v2 medium (n=1868) | assoc n10v2 hard (n=388) | assoc n10v2 all (n=3940) |
| 7b rl inte | 48.337 | **34.904** | **25.258** | 39.695 |
| 7b sft rs32 inte | **67.043** | 24.732 | 17.784 | **42.132** |
| 7b rl coun | 55.641 | 31.156 | **23.454** | **40.863** |
| 7b sft rs32 coun | 43.943 | 22.645 | 15.464 | 31.041 |
| 7b cot init | 29.513 | 11.135 | 6.701 | 18.553 |
| 7b dp sft inte | 56.116 | 18.094 | **21.907** | 34.721 |
| 7b dp sft coun | 31.116 | 18.737 | 18.299 | 23.985 |
| 7b dp init | 7.898 | 2.677 | 1.546 | 4.797 |
| | counte n10v2 easy (n=24) | counte n10v2 medium (n=457) | counte n10v2 hard (n=231) | counte n10v2 all (n=712) |
| 7b rl inte | **0.000** | **13.567** | **11.688** | 12.500 |
| 7b sft rs32 inte | **4.167** | 11.597 | **10.390** | 10.955 |
| 7b rl asso | **4.167** | **14.880** | **15.584** | **14.747** |
| 7b sft rs32 asso | **4.167** | 10.503 | 7.359 | 9.270 |
| 7b cot init | **4.167** | 8.753 | 4.762 | 7.303 |
| 7b dp sft inte | **8.333** | 6.127 | 6.494 | 6.320 |
| 7b dp sft asso | **8.333** | 9.409 | **9.957** | 9.551 |
| 7b dp init | **4.167** | 1.751 | 0.433 | 1.404 |

Table 9: Across-level generalization of 7B models fine-tuned with RLVR, SFT with reasoning chains, and SFT with direct prediction (test, filtered). Row specify which level trained on, column specify which level evaluated on. System accuracy (average CORRECT, see eq. (3)) on evaluation sets of different difficulties, as measured by $|V_{\text{rel}}|$, the *size of the relevant subgraph* to the query variable. Note that difficulty is not comparable across different levels. The models are trained on a mix of easy/medium/hard questions. Systems not significantly worse than the best overall (with a monte-carlo paired permutation test with n=10000) are bolded.

*Across-level generalization* of SFT on rejection-sampled reasoning chains (RS32) across three levels are included table 9. Comparing RS32 with COT INIT shows that fine-tuning on correct reasoning chains improves the performance of the LLM when evaluated on a different level from training. Comparing RS32 with RL shows that it underperforms RLVR on across-level generalization except slightly on the intervention to association direction, and again the gap is especially large on more difficult queries. RS32 outperforms direction prediction SFT (DP) on average on intervention levels and association levels. Overall, these results show that fine-tuning on reasoning-chains and the on-policy nature of RLVR both contribute to its superior performance on across-level generalization.

### D.6 DETERMINISTIC COUNTERFACTUAL PROBLEMS

**Deterministic Counterfactual Problems**   In our main experiment (section 4.2, fig. 3), we found that no method generalizes reliably on the counterfactual level. Our counterfactual level problems require abduction and marginalization, which is challenging for the LLMs even at small graph sizes. To understand the limitations of LLMs on this level better, we perform an evaluation of the fine-tuned models on a set of *simpler* counterfactual queries that are have *small* graphical models with *deterministic* mechanisms that remove the need of marginalization. See fig. 26 for an example problem.

---

**User Prompt Deterministic Counterfactual**

Here's a structural causal model over discrete random variables. The Variables are v0, v1, v2. Here are the Values they can take on.

v0 can take values in [0, 1]
v1 can take values in [0, 1]
v2 can take values in [0, 1]

Here's the causal directed acyclic graph (DAG):
strict digraph {
v0;
v1;
v2;
v0 → v1;
v2 → v1;
}

Here are the causal conditional probability tables (CPT) associated with the DAG:
CPTs for v2:
P(v2) = [0.71, 0.29]

CPTs for v0:
P(v0) = [0.08, 0.92]

CPTs for v1:
P(v1 | v2=0,v0=0) = [1, 0]
P(v1 | v2=0,v0=1) = [0, 1]
P(v1 | v2=1,v0=0) = [0, 1]
P(v1 | v2=1,v0=1) = [0, 1]

Furthermore, each variable v is assumed to depend deterministically on its parents pa(v) and a collection of independent exogenous selector variables, one for each possible joint assignment to pa(v), whose marginal distribution is defined to be p(v | pa(v)). Given a particular assignment to pa(v), v takes on the value of the selector variable corresponding to that particular assignment pa(v).

Here's your Question: What is the marginal distribution of v1 given we first observed v2 = 0 and then intervened to set v0 to 0?

———-

Now start your solution process. Be precise.

---

Figure 26: Example user prompt $x_{\text{user}}$ containing causal graph and query converted from CLadder (Jin et al., 2023) deterministic counterfactual subset. The original question was "We know that blowing out the candle or candle with wax causes dark room. We observed the candle is out of wax. Would the room is dark if not blowing out the candle instead of blowing out the candle?" (question ID 9538). The answer is "no", or $[1, 0]$ in our format.

Specifically, we construct the evaluation data by selecting a subset of problems from the "deterministic counterfactual" subset of the CLadder dataset (Jin et al., 2023). The det-counterfactual subset consists of 1422 counterfactual queries on small graphs (3 to 4 endogenous nodes) with deterministic mechanisms. We keep only questions that have exactly one observation and action (e.g. $p(Y_{(x=0)} \mid z = 1)$) to match the setting of our training and test sets. This leaves us with 790 problems. Instead of the natural language format, we represent the problems as formal problems, to match our training and test sets.

Comparing RLVR and SFT models in the deterministic setting (table 10 left), we find that LLMs fine-tuned with RLVR (with reasoning) are significantly better than those fine-tuned with SFT (with direct prediction), which is opposite to the ordering of models on our more challenging counterfactual test set. Overall, regardless of the fine-tuning method, LLMs also generally perform much better on the simpler setting of **deterministic** counterfactuals (table 10 left) than our counterfactuals that

require abduction and marginalization (table 10 right). This contrast suggests that the requirement to perform nontrivial abduction followed by marginalization is a major challenge of the counterfactual problems in our dataset.

| | cladder det-counterfactual (n=790) | | counte n10v2 (n=24) |
|---|---|---|---|
| rl 32b | **99.747** | rl 32b | 4.167 |
| rl 32b curriculum | **99.747** | rl 32b curriculum | 0.000 |
| rl 32b init | 98.987 | rl 32b init | 0.000 |
| rl 7b | 84.937 | rl 7b | 4.167 |
| rl 7b curriculum | 84.937 | rl 7b curriculum | 0.000 |
| rl 7b init | 67.468 | rl 7b init | 4.167 |
| rl 3b | 64.051 | rl 3b | 4.167 |
| rl 3b curriculum | 57.722 | rl 3b curriculum | 0.000 |
| rl 3b init | 40.506 | rl 3b init | 0.000 |
| sft 32b | 70.633 | sft 32b | **29.167** |
| sft 32b init | 81.392 | sft 32b init | 4.167 |
| sft 7b | 61.139 | sft 7b | **20.833** |
| sft 7b init | 43.038 | sft 7b init | 4.167 |
| sft 3b | 48.734 | sft 3b | 0.000 |
| sft 3b init | 47.089 | sft 3b init | 0.000 |

Table 10: Performance of models trained with our counterfactual training split, and tested on the CLadder (Jin et al., 2023) det-counterfactual subsplit which consist of counterfactual queries on small graphs with deterministic mechanisms. We include queries with exactly one observation and action (790 out of 1422), and represent the questions formally rather than in natural language (in CLadder) to match our setting. Systems not significantly worse than the best (with a monte-carlo paired permutation test with n=10000) are bolded. [9]

### D.7 COPY AND ARITHMETIC ERRORS

**Copy and Arithmetic Error Analysis** In section 4.3 (Analysis-III), we use LLMs to analyze the reasoning trace of fine-tuned models, and categorize their high level marginalization strategy as well as detect probability derivation errors (e.g. falsely assuming independence among variables, missing terms in marginalization formulas). To have a more complete understanding of the error patterns, we extend the LLM-aided error analysis to *copy errors* and *arithmetic errors*. Copy errors are errors where probability values are copied incorrectly from the problem statement (e.g. substituting an incorrect value for a term like $p(v_1 = 0 \mid v_2 = 1)$ during calculation). Arithmetic errors are numerical errors in addition, subtraction, multiplication, or division. We use o4-mini to analyze on the reasoning traces, using prompts in fig. 12 to detect copy and arithmetic errors respectively. The first author checked 10 sample annotations per category for copy (has errors, no errors, not applicable) and arithmetic errors (has errors, no errors, not applicable) and found that they agreed with the annotation 25/30 and 23/30 respectively for copy error and arithemtic error.

The analysis of RLVR traces are visualized in fig. 27. First, fig. 27 (top) show that copy errors generally decrease after training as expected. We also see that for 7b and 32b models and especially on the association and counterfactual levels, examples without copy errors (pink) are nonetheless often incorrect (pink with shade), suggesting these examples suffer other kinds of errors that lead to wrong answers. Next, fig. 27 (bottom) show that arithmetic errors reduce but not by a large magnitude after fine-tuning. We also see that for 7b and 32b models on intervention and association levels, traces with arithmetic errors (green) are nonetheless often marked correct (green without shade). This is likely due to our correctness function (used during training and

---

[9]We re-ran the rl 7b training due to a lost checkpoint, and report the cladder det-counterfactual performance of this new run.

evaluation) check answers rounded to the nearest $0.01$, which allows for slight numerical errors.

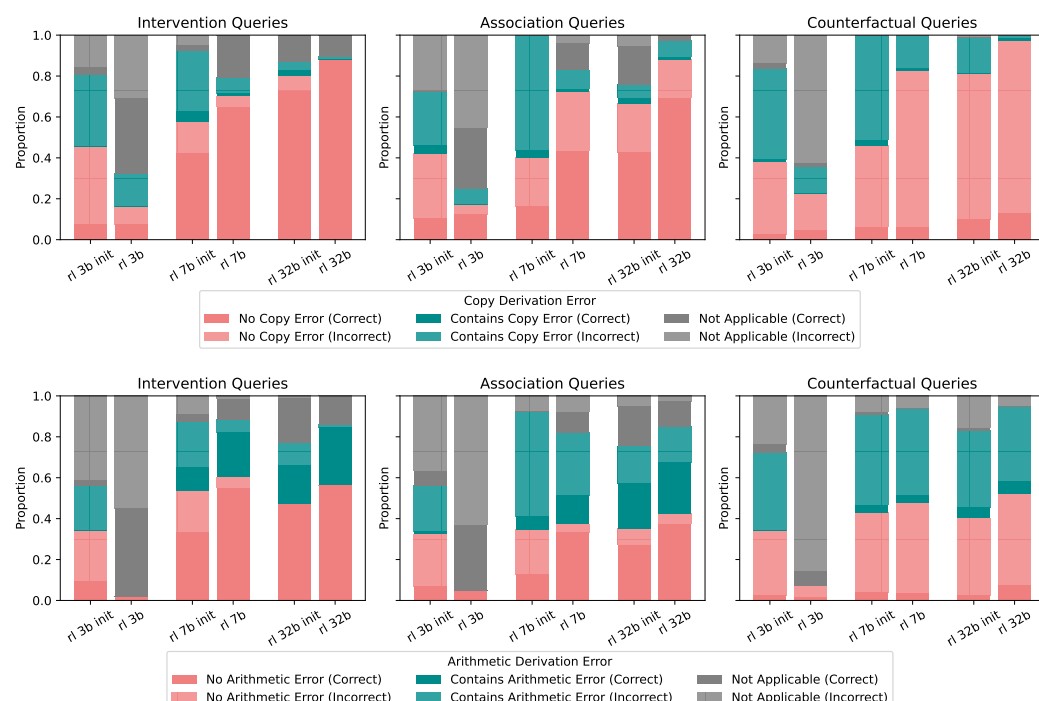

Figure 27: LLM judge (o4-mini) analysis of copy errors (top) and arithmetic errors (bottom) before and after RLVR, on 80 samples per level. Judge prompts (including category definitions) are included in fig. 12.

## D.8 ERROR IMPROVEMENT OF RLVR VS. SFT WITH REASONING CHAINS

**Error improvement of RLVR vs. SFT with Reasoning Chains**    In fig. 28, we compare the reduction of various kinds of errors achieved by RLVR versus SFT *with reasoning chains* based on 7B models (see appendix D.5 for details about the setup), and find that RLVR is generally more effective at reducing all of probability, copying, and arithmetic errors, and especially so for probability derivation errors.

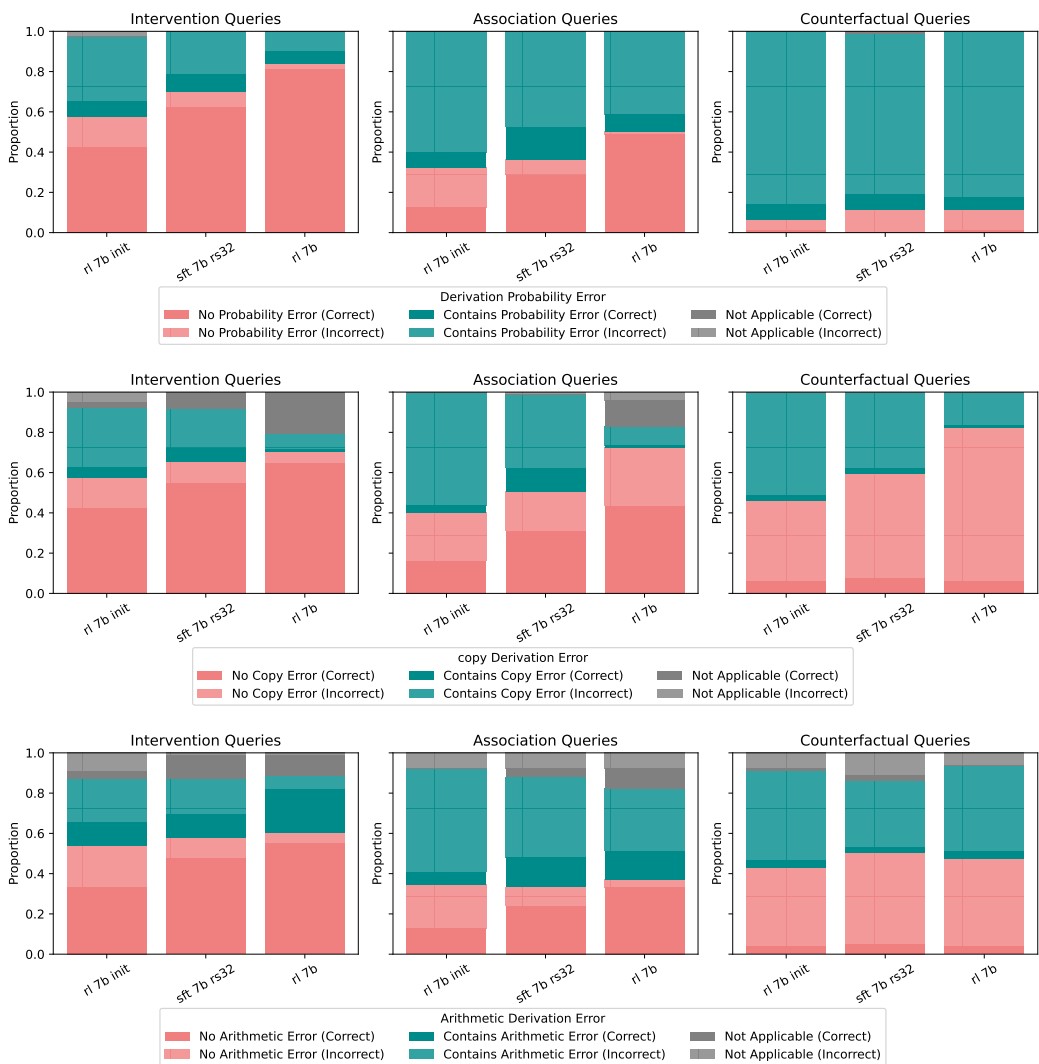

Figure 28: LLM judge (o4-mini) analysis of derivation (top), copy errors (mid), and arithmetic errors (bottom) before and fine-tuning a 7B LLM, on 80 samples per level. Judge prompts (including category definitions) are included in fig. 11 and fig. 12 .

## D.9    3B RLVR WITH MORE TRAINING STEPS

**Training 3B models for longer with RLVR to match 7B and 32B compute leads to only minor improvements and no qualitative change in marginalization strategy.**    We trained 3B models for 7500 steps in our main experiments, which uses about a quarter of the compute compared to 7B and 32B training. We perform an experiment extending their training steps to 30000 steps to match the compute. The results are included in table 11, we can see the additional steps lead to minor improvements in accuracy across levels. In fig. 29, we also compare the marginalization strategy of 3B model trained with 30000 steps of RLVR versus those trained with 7500 steps and find little qualitative difference.

| | interv n10v2 easy (n=1086) | interv n10v2 medium (n=664) | interv n10v2 hard (n=348) | interv n10v2 all (n=2098) | assoc n10v2 easy (n=1684) | assoc n10v2 medium (n=1868) | assoc n10v2 hard (n=388) | assoc n10v2 all (n=3940) | counte n10v2 easy (n=24) | counte n10v2 medium (n=457) | counte n10v2 hard (n=231) | counte n10v2 all (n=712) |
|---|---|---|---|---|---|---|---|---|---|---|---|---|
| 3b rl 7.5k | 89.779 | **8.584** | 5.172 | 50.048 | 47.268 | **10.278** | **12.887** | 26.345 | **4.167** | **7.221** | 6.926 | **7.022** |
| 3b rl 30k | **93.002** | **8.886** | **6.322** | **52.002** | **53.860** | **10.600** | **12.113** | **29.239** | **4.167** | **7.002** | **8.658** | **7.444** |

Table 11: Within level generalization (test, filtered) of 3b models fine-tuned with rl for 7500 and 30000 steps. System accuracy (average CORRECT, see eq. (3)) when training and evaluating on queries from same level. Stratified by query level, and difficulty within each level, as measured by $|V_{\mathrm{rel}}|$, the *size of the relevant subgraph* to the query variable. Note that difficulty is not comparable across different levels. The models are trained on a mix of small/medium/large questions. Systems not significantly worse than the best (with a monte-carlo paired permutation test with n=10000) are bolded.

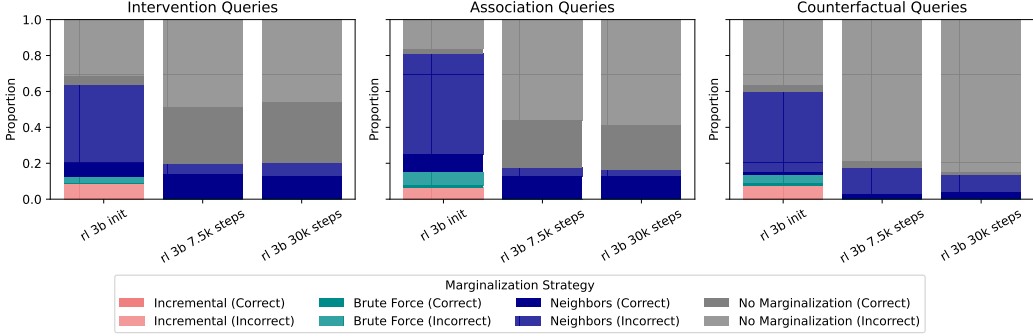

Figure 29: LLM judge (o4-mini) analysis of reasoning strategy of 3B RLVR trained with 7.5k steps and 30k steps, on 80 samples per level. Judge prompts (including category definitions) are included in fig. 11.