# OpenReview forum: "Generalization of RLVR Using Causal Reasoning as a Testbed"
_ICLR.cc/2026/Conference — ICLR 2026 Poster_

### Official Review · Reviewer_Cf3X · 2025-10-25

**Soundness:** 3
**Presentation:** 1
**Contribution:** 2
**Rating:** 4
**Confidence:** 5

**Summary:**

This paper studies the generalization of reinforcement learning with verifiable rewards (RLVR). The authors selected causal reasoning as the probing task. Specifically, authors generate a benchmark of causal graphs alongside corresponding questions. With respect to the experiments, the authors make a comparison between SFT and RLVR and find that RLVR outperforms SFT on some subsets of the benchmark. Finally, the authors conclude that RLVR indeed enhances the generalization abilities of causal reasoning at the association and intervention level.

**Strengths:**

1. The experiments are comprehensive and sufficient to verify the authors' claims.

2. The analysis is comprehensive, authors provide in-depth analyses.

**Weaknesses:**

1. I think the first weakness lies in the writing of the abstract. The current version seems a bit colloquial rather than a formal academic paper. Specifically, the sentence "We choose this setting because causality is an important area that LLMs still struggle with, and because this setting ..." is too colloquial and lengthy. I suggest authors consider rewriting the abstract thoroughly and consider using shorter sentences.

2. Similarly, in the introduction, the sentence "However, we focus on identifying situations in which RLVR itself generalizes effectively
(versus not), and we focus on the causal reasoning domain." may be informal. Authors employ "we focus on" twice and "(versus not)" may be a little informal. Besides, "in which RLVR itself generalizes effectively" can be confusing, what dose RLVR itself mean here? I would encourage authors to revise the introduction throughly. For example, "our work differs from prior studies by focusing on an essential and challenging task: causal reasoning."

3. I would suggest that authors include a discussion on the practical value of the studied formal causal reasoning. Since I believe LLMs are more targeted at the commonsense setting, and there already exist lots of formal causal tools in the area of causal inference.

4. Authors should consider assigning a formal name for their datasets (e.g., RLCausal or other names), since it is an important contribution of this work.

5. As there already exist other formal causal reasoning benchmarks (e.g., the CLADDER [1]), I would suggest that authors add an individual section on the differences between their newly proposed datasets and existing benchmarks. Why can't other benchmarks test the generalization abilities of RLVR?

6. The colors in Figure 1 are too light; the authors should consider deepening them. The text is too small, and it's unnecessary to show every detail of the sample; simplifying the content would allow for a larger font size. Besides, I do not quite get the main structure of the Task Formulation in this version. Authors should revise Figure 1 to a more abstract and summarized representation.

> [1] Jin Z, Chen Y, Leeb F, et al. Cladder: Assessing causal reasoning in language models[J]. Advances in Neural Information Processing Systems, 2023, 36: 31038-31065.

Happy to raise the score if the authors could address my concerns.

**Questions:**

Please refer to the weaknesses part.

---

> ### Author Response · Authors · 2025-11-24
> **Response to W1 and W2**
>
> We thank the reviewer for the insightful feedback and questions. We are encouraged that you found our experiments comprehensive and claims sufficiently supported, and our analysis deep and comprehensive.
>
> We address each of the weaknesses below:
> > **(W1)** W1: I think the first weakness lies in the writing of the abstract. The current version seems a bit colloquial rather than a formal academic paper. Specifically, the sentence "We choose this setting because causality is an important area that LLMs still struggle with, and because this setting ..." is too colloquial and lengthy. I suggest authors consider rewriting the abstract thoroughly and consider using shorter sentences.
>
> Thank you for the helpful feedback on our abstract. We have rewritten our abstract for concision, formality, and clarity. We include it below:
>
> > Reinforcement learning with verifiable rewards (RLVR) has emerged as a promising paradigm for post-training large language models (LLMs) on complex reasoning tasks. Yet, the conditions under which RLVR yields robust generalization remain poorly understood. This paper provides an empirical study of RLVR generalization in the setting of probabilistic inference over causal graphical models. This setting offers two natural axes along which to examine generalization: (i) the level of the probabilistic query---associational, interventional, or counterfactual---and (ii) the structural complexity of the query, measured by the size of its relevant subgraph. We construct datasets of causal graphs and queries spanning these difficulty axes and fine-tune Qwen-2.5-Instruct models using RLVR or supervised fine-tuning (SFT). We vary both the model scale (3B–32B) and the query level included in training. We find that RLVR yields stronger within-level and across-level generalization than SFT, but only for specific combinations of model size and training query level. Further analysis shows that RLVR’s effectiveness depends on the model’s initial reasoning competence. With sufficient initial competence, RLVR improves an LLM’s marginalization strategy and reduces errors in intermediate probability calculations, producing substantial accuracy gains, particularly on more complex queries.  These findings show that RLVR can improve specific causal reasoning subskills, with its benefits emerging only when the model has sufficient initial competence.
>
> > **(W2)** W2: Similarly, in the introduction, the sentence "However, we focus on identifying situations in which RLVR itself generalizes effectively (versus not), and we focus on the causal reasoning domain." may be informal. Authors employ "we focus on" twice and "(versus not)" may be a little informal. Besides, "in which RLVR itself generalizes effectively" can be confusing, what dose RLVR itself mean here? I would encourage authors to revise the introduction thoroughly. For example, "our work differs from prior studies by focusing on an essential and challenging task: causal reasoning."
>
> Thank you for the helpful feedback on our introduction. We have rewritten our introduction and uploaded our revised draft. We revised for formality and clarity overall, and content-wise focused on adding a clearer motivation of our data and task setup by comparing with CLadder [1], which we include below.
>
> > Causal inference provides a structured setting for examining RLVR generalization, because its three levels of inference---associational, interventional, and counterfactual, known collectively as the causal ladder [1, 2] ---form a hierarchy that supports both within- and across-level generalization tests. CLadder [3] covers this hierarchy and serves as a holistic causal inference benchmark, requiring models to interpret natural-language scenarios, identify query types, and formalize causal expressions, in addition to performing the derivation and calculation needed to answer the query. In our setting, we focus on the derivation and calculation components, which become more involved as graphs grow larger and more interconnected, making them useful for probing how LLMs fine-tuned with RLVR handle questions that require an increasing number of reasoning steps. We therefore construct our own dataset RLCausal, whose questions include fully specified causal graphs instead of natural language scenarios, and vary the graph structure, extending beyond CLadder’s manually curated topologies to larger, randomly generated 10-node graphs.
>
> [1] E. Bareinboim, J. D. Correa, D. Ibeling, and T. F. Icard, ‘On Pearl’s Hierarchy and the Foundations of Causal Inference’, Probabilistic and Causal Inference, 2022.
>
> [2] J. Pearl and D. Mackenzie, The Book of Why: The New Science of Cause and Effect, 1st edn. USA: Basic Books, Inc., 2018.
>
> [3] Z. Jin et al., ‘CLadder: A Benchmark to Assess Causal Reasoning Capabilities of Language Models’, in Thirty-seventh Conference on Neural Information Processing Systems, 2023.

---

> ### Author Response · Authors · 2025-11-24
> **Response to W3 and W4**
>
> > **(W3)** W3: I would suggest that authors include a discussion on the practical value of the studied formal causal reasoning. Since I believe LLMs are more targeted at the commonsense setting, and there already exist lots of formal causal tools in the area of causal inference.
>
> Thank you for the useful suggestion. We added a paragraph to our conclusion and future work section discussing the practical usefulness of our task and data, beyond our findings. We note that because our focus is on analyzing RLVR’s generalization, it benefited us to select a task that can be solved by existing causal tools, so that we can generate training and evaluation data systematically and scalably, much like CLadder [3]. However, our focus on RLVR, which is used to train LLMs to do extended step-by-step reasoning, pushed us to deviate from the natural language understanding aspects of CLadder’s task. Instead, we focus purely on the derivation and calculation subskills, which we believe are better probing tasks for step-by-step reasoning ability, since the derivation and calculation subtasks become much more involved as we scale to larger and more complicated graphs). This ultimately led to our task of choice of abstract formal inference in causal graphs. In practice, one may prefer to solve such problems with a causal tool instead of a LLM. However, such a task can still train useful subskills that might benefit LLMs, such as refining their ability to apply abstract knowledge of probability and causality to derive formulas, and refining their lower level copying and arithmetic skills.
>
> We include our added discussion below:
>
> > Recent work has started building domain-specific task suites for specializing LLMs to solve complex tasks in real-world science and engineering domains via RLVR fine-tuning. For example, Biomni-R0 [4] uses RLVR to fine-tune LLMs on a range of biomedical tasks that requires step-by-step reasoning, and ether0 [5] does so similarly for chemistry. Beyond supporting our analysis of RLVR generalization, our dataset can be used to train and probe sub-skills (e.g. organizing marginalization of latent variables, applying probability identities, performing arithmetic) that are useful to solving more practical causal and probabilistic problems, even though our problems are abstracted away from real-world scenarios. Our data generation process can also be configured to scale up the number of problems as well as their difficulty, by increasing graph size and the cardinality of each variable as appropriate.
>
> > **(W4)** W4: Authors should consider assigning a formal name for their datasets (e.g., RLCausal or other names), since it is an important contribution of this work.
>
> Thank you for the suggestion. We will consider RLCausal as a name for our dataset.
>
> [3] Z. Jin et al., ‘CLadder: A Benchmark to Assess Causal Reasoning Capabilities of Language Models’, in Thirty-seventh Conference on Neural Information Processing Systems, 2023.
>
> [4] Biomni and T. Sky RL, ‘Biomni-R0: Using RL to Hill-Climb Biomedical Reasoning Agents to Expert-Level’, Biomni - A General-Purpose Biomedical AI Agent. Sept-2025.
>
> [5] S. M. Narayanan et al., ‘Training a Scientific Reasoning Model for Chemistry’, arXiv [cs.LG]. 2025.

---

> ### Author Response · Authors · 2025-11-24
> **Response to W5**
>
> > **(W5)** W5: As there already exist other formal causal reasoning benchmarks (e.g., the CLadder [3]), I would suggest that authors add an individual section on the differences between their newly proposed datasets and existing benchmarks. Why can't other benchmarks test the generalization abilities of RLVR?
>
> Thank you for urging us to clarify this important question. We added a discussion in the introduction section focusing on CLadder [3], and expanded our related works section on LLM for Causality to discuss differences with other benchmarks and why they are not as suitable as our dataset for studying RLVR. The main feature of our task/dataset is focus on the abstract and formal step-by-step reasoning across the causal hierarchy (which is partially shared by CLadder but not other datasets that focus more on evaluating causal knowledge about the real world), and a focus on scaling up the problem difficulty by introducing more diverse graphs and larger graphs (which is not shared by CLadder).
>
> We added the following to our introduction (repeated from W3 for readability).
>
> > Recent work has started building domain-specific task suites for specializing LLMs to solve complex tasks in real-world science and engineering domains via RLVR fine-tuning. For example, Biomni-R0 [4] uses RLVR to fine-tune LLMs on a range of biomedical tasks that requires step-by-step reasoning, and ether0 [5] does so similarly for chemistry. Beyond supporting our analysis of RLVR generalization, our dataset can be used to train and evaluate sub-skills (e.g. organizing marginalization of latent variables, applying probability identities, performing arithmetic) that are useful to solving more practical causal and probabilistic problems through RLVR, even though our problems are abstracted away from real-world scenarios. Our data generation process can also be configured to scale up the number of problems as well as their difficulty, by increasing graph size and the cardinality of each variable as appropriate.
>
> We added the following to our related works section.
>
> > LLMs' understanding of causality has been receiving increasing interest [6], as LLMs make their way into domains like medicine and law where causal reasoning is essential. One line of work investigates the real world causal knowledge and reasoning of pretrained LLMs [7,8,9]. These benchmarks are valuable for assessing whether LLMs encode plausible causal facts about the world, but they are less suited for probing RLVR, which focuses more on step-by-step reasoning over such facts to draw new conclusions, a separate skill from memorization of the facts or conclusions. A line of datasets that focus more on formal causal reasoning capabilities include CLadder [3], Corr2Cause [10], and Carl-gt [11]. Among these more formal benchmarks, the closest to ours is CLadder [3], whose questions and answers are algorithmically generated by running a causal-inference engine over structured graphical models and expressed as natural language scenarios spanning all rungs of the causal ladder.  Similar to CLadder, our benchmark is built on synthetic casual graphical models, focusing more on reasoning over memorization. Different from CLadder, our benchmark isolates causal reasoning from natural language understanding by providing a full specification of the abstract causal graphic in the question rather than converting it to a natural language scenario. We also introduce substantially more structural diversity by using larger graphs (10 nodes) and a wider range of randomly generated graph topologies (80 distinct for training and 200 distinct for dev/test). These two features make our dataset a focused and challenging dataset suitable for studying within-level and cross-level generalization of RLVR.

---

> ### Author Response · Authors · 2025-11-24
> **Response to W5 Continued**
>
> [3] Z. Jin et al., ‘CLadder: A Benchmark to Assess Causal Reasoning Capabilities of Language Models’, in Thirty-seventh Conference on Neural Information Processing Systems, 2023.
>
> [4] Biomni and T. Sky RL, ‘Biomni-R0: Using RL to Hill-Climb Biomedical Reasoning Agents to Expert-Level’, Biomni - A General-Purpose Biomedical AI Agent. Sept-2025.
>
> [5] S. M. Narayanan et al., ‘Training a Scientific Reasoning Model for Chemistry’, arXiv [cs.LG]. 2025.
>
> [6] L. Yu et al., ‘CausalEval: Towards Better Causal Reasoning in Language Models’, in Proceedings of the 2025 Conference of the Nations of the Americas Chapter of the Association for Computational Linguistics: Human Language Technologies (Volume 1: Long Papers), 2025, pp. 12512–12540.
>
> [7] E. Kiciman, R. Ness, A. Sharma, and C. Tan, ‘Causal Reasoning and Large Language Models: Opening a New Frontier for Causality’, Transactions on Machine Learning Research, 2024.
>
> [8] M. Zečević, M. Willig, D. S. Dhami, and K. Kersting, ‘Causal Parrots: Large Language Models May Talk Causality But Are Not Causal’, Transactions on Machine Learning Research, 2023.
>
> [9] H. Chi et al., ‘Unveiling Causal Reasoning in Large Language Models: Reality or Mirage?’, in The Thirty-eighth Annual Conference on Neural Information Processing Systems, 2024.
>
> [10] Z. Jin et al., ‘Can Large Language Models Infer Causation from Correlation?’, in The Twelfth International Conference on Learning Representations, 2024.
>
> [11] R. Tu, H. Kjellström, G. E. Henter, and C. Zhang, ‘CARL-GT: Evaluating Causal Reasoning Capabilities of Large Language Models’, arXiv [cs.CL]. 2024.

---

> ### Author Response · Authors · 2025-11-24
> **Response to W6**
>
> > **(W6)** W6: The colors in Figure 1 are too light; the authors should consider deepening them. The text is too small, and it's unnecessary to show every detail of the sample; simplifying the content would allow for a larger font size. Besides, I do not quite get the main structure of the Task Formulation in this version. Authors should revise Figure 1 to a more abstract and summarized representation.
>
> Thank you for the informative feedback and suggestions! We have increased the saturation of our figure 1, changed the color palette, increased font size and abstracted away details to improve clarity and readability.

---

> > ### Comment · Reviewer_Cf3X · 2025-11-24
> > **Response to rebuttal**
> >
> > Thank the authors for their detailed response. Most of my concerns have been addressed.
> > I have increased my score to a borderline accept.

---

### Official Review · Reviewer_5AJc · 2025-10-28

**Soundness:** 3
**Presentation:** 3
**Contribution:** 3
**Rating:** 6
**Confidence:** 4

**Summary:**

The paper investigates the relationship between large language models’ (LLMs) post-training mechanisms and their ability to perform causal reasoning tasks. The author introduces a data generation process to create and evaluate causal question-answering scenarios. The study then compares models post-trained using Supervised Fine-Tuning (SFT) and Reinforcement Learning from Verbal Reward (RLVR, including GRRO and DAPO variants) across different evaluation dimensions.

**Strengths:**

The paper is well-structured and clearly written, with a logical flow that makes it easy to follow the author’s reasoning. The analysis sections are particularly strong: insightful, well-grounded, and supported by detailed experiments. The results are extensive and could serve as a valuable reference for future researchers studying the intersection of LLM post-training and causal reasoning.
I was initially debating between a rating of 6 and 8 and currently lean toward the former, though I remain open to adjusting this based on the rebuttal and other reviewers’ feedback.

**Weaknesses:**

1. The RLVR experiments use 7.5K and 2.5K samples, while the SFT model is trained on 5K samples. This discrepancy makes the quantitative comparison between models less reliable. I suggest adding an ablation study where models (or checkpoints) are trained on the same amount of data and for comparable GPU hours to mitigate this concern.

2. LLMs learn differently from humans as they rely primarily on language pattern recognition rather than true causal inference. The proposed causal reasoning tasks implicitly require two skills: (a) retrieving or identifying the correct numerical values from context, and (b) performing basic computations. It would strengthen the paper to include a detailed error analysis comparing SFT and RLVR models on these sub-tasks, in addition to the LLM-judge results that emphasize human-like problem-solving performance.

**Questions:**

N/A

---

> ### Author Response · Authors · 2025-11-24
> **Response to W1**
>
> We thank the reviewer for the thoughtful feedback and suggestions! We are encouraged that you found our presentation clear and logical, our results valuable and extensive and that our findings insightful and well-supported.
>
> We address each of the weakness below:
>
> > **(W1)** W1: The RLVR experiments use 7.5K and 2.5K samples, while the SFT model is trained on 5K samples. This discrepancy makes the quantitative comparison between models less reliable. I suggest adding an ablation study where models (or checkpoints) are trained on the same amount of data and for comparable GPU hours to mitigate this concern.
>
> Thank you for raising this question. We’d like to clarify that the 7.5K, 2.5K, and 5K refer to the number of gradient updates rather than the number of samples. All models (SFT or RL) are trained on the same 8000 training examples. The 5K steps of SFT correspond to five epochs on the training data at which we already observe the U-shape overfitting curve on dev loss, and we choose the best earlier checkpoint by dev loss. The 7.5K to 2.5K step ratio of RL experiments was already chosen to control for GPU hours between 7B and 32B RL training runs (120-130 hours on an 8-gpu machine), which we discussed in the “Fine-tuning And Inference” Setup paragraph of Sec. 4. Our 3B RL experiment on the other hand uses 7.5K steps, which turns out to be a quarter of the gpu hours (about 30 hours) used by 7B and 32B runs. However at 7.5K steps we observed already that its improvement has slowed down significantly, and so we decided to stop and spend our compute on the larger experiments instead. To make sure that our findings about the 3B models are robust to it receiving more training steps, we extend the training of 3B RL models to 30K steps, which matches the compute of 7B and 32B RL models. We discuss the results in Sec. D.9 and also include a summary below.
>
> Setup. Train 3B RL models for another additional 22.5K steps to reach 30K total steps, which matches the 120 to 130 hour compute used to train 7B and 32B models.
>
> Results. 3B models received a slight improvement in accuracy across the board, but the quadrupling of training did not substantially change its main qualitative feature: its marginalization strategy remains stuck at directly predicting the answer rather than performing step by step marginalization.
>
> Conclusion. While the numerical results improved slightly from 7.5K steps of training to 30K steps, our findings about the 3B model’s reasoning behavior were not significantly impacted.
>
> | Model | interv n10v2 easy (n=1086) | interv n10v2 medium (n=664) | interv n10v2 hard (n=348) | interv n10v2 all (n=2098) | assoc n10v2 easy (n=1684) | assoc n10v2 medium (n=1868) | assoc n10v2 hard (n=388) | assoc n10v2 all (n=3940) | counte n10v2 easy (n=24) | counte n10v2 medium (n=457) | counte n10v2 hard (n=231) | counte n10v2 all (n=712) |
> | --- | --- | --- | --- | --- | --- | --- | --- | --- | --- | --- | --- | --- |
> | 3b rl 7.5k step | 89.779 | **8.584** | **5.172** | 50.048 | 47.268 | **10.278** | **12.887** | 26.345 | **4.167** | **7.221** | **6.926** | **7.022** |
> | 3b rl 30k step | **93.002** | **8.886** | **6.322** | **52.002** | **53.860** | **10.600** | **12.113** | **29.239** | **4.167** | **7.002** | **8.658** | **7.444** |

---

> ### Author Response · Authors · 2025-11-24
> **Response to W2**
>
> > **(W2)** W2: LLMs learn differently from humans as they rely primarily on language pattern recognition rather than true causal inference. The proposed causal reasoning tasks implicitly require two skills: (a) retrieving or identifying the correct numerical values from context, and (b) performing basic computations. It would strengthen the paper to include a detailed error analysis comparing SFT and RLVR models on these sub-tasks, in addition to the LLM-judge results that emphasize human-like problem-solving performance.
>
> Thank you for raising this question. We agree it is useful to also analyze the lower level skills of copying correct numerical values from question statements and performing basic arithmetic. We added an error analysis focusing on copy and arithmetic errors in Sec. D.7 and Sec. D.8 and include a summary below.
>
> Setup. Use o4-mini to analyze the copy errors and arithmetic errors similar to how we analyzed the marginalization strategy and probability derivation errors. Direct prediction SFT do not have reasoning chains and thus are not comparable to RLVR in this analysis, we thus collect a rejection-sampled SFT dataset and fine-tune on it.
>
> Findings. Copy errors generally decrease after RLVR, but many 7B/32B traces without copy errors are still wrong, especially on association and counterfactual queries, suggesting other reasoning errors remain. Arithmetic errors also decrease slightly after RLVR, but not dramatically. For 7B/32B on association/intervention, many traces with arithmetic errors are still judged correct, likely because our correctness check is tolerant to small rounding differences (answers rounded to 0.01). SFT on rejection-sampled reasoning chains reduces all types of errors compared to initialization but reduces errors less effectively compared to RLVR.

---

> > ### Comment · Reviewer_5AJc · 2025-11-24
> >
> > The authors' rebuttal is comprehensive, and my comments are addressed well. I have raised my evaluation accordingly.

---

### Official Review · Reviewer_yW6b · 2025-11-04

**Soundness:** 3
**Presentation:** 3
**Contribution:** 3
**Rating:** 4
**Confidence:** 3

**Summary:**

This paper investigates the generalization capabilities of RLVR for post-training LLMs, using probabilistic inference in causal graphical models as a testbed. The authors fine-tune Qwen-2.5-Instruct models (3B-32B parameters) using both RLVR and SFT on datasets of causal graphs and queries spanning three difficulty levels and complexity. The work contributes to understanding the conditions under which RLVR effectively generalizes, highlighting both its strengths and limitations in challenging formal reasoning tasks.

**Strengths:**

- The choice of probabilistic inference in causal graphical models as a testbed is genuinely innovative. Unlike prior RLVR generalization studies that focus on text/visual reasoning tasks, this formal mathematical domain enables precise control and analysis.
- The findings have practical implications: practitioners should check if their base model has sufficient reasoning capability before investing in RLVR. The identification that counterfactual reasoning remains unsolved even with RLVR and 32B models highlights a key challenge for the field.

**Weaknesses:**

- SFT is trained only to predict final answers while RLVR generates full reasoning chains. This creates an asymmetric comparison that conflates two factors: (1) reasoning vs. direct prediction and (2) RL vs. supervised learning. A fair strategy is to include an SFT baseline trained on optimal reasoning chains (generated by the solver or sampled from successful RLVR rollouts). This would isolate whether gains come from RL exploration or simply having reasoning chains.
- The paper observes 3B models fail to benefit from RLVR and regress to direct prediction after training, but doesn't investigate what specifically these models lack.
- The reward is simply r = 0.8 · accuracy + 0.2 · format with threshold t=0.01. Is the 0.8/0.2 weighting optimal? Would shaped rewards (partial credit for intermediate steps) help?
- The writting is vague. For example, the description of the third finding "What did RLVR learn? " is too abstract, containing too many technical terms. They should present this result in a concrete style.
- Why not compare RLVR with other RL-based post-training paradigm, such as RLHF and RLAIF?

**Questions:**

- Can you provide results for SFT trained on reasoning chains (even if just for 7B/one level)?
- Can models solve trivially simple counterfactuals (e.g., 3-node graphs, no marginalization needed)?

---

> ### Author Response · Authors · 2025-11-24
> **Response to W1 + Q1**
>
> We thank the reviewer for the informative feedback and suggestions! We are encouraged that you found our causal-graphical-model testbed innovative and appreciate your comments on the value of our findings for LLM practitioners.
>
> We address each of the weaknesses / questions below:
>
> > **(W1 + Q1)** W1: SFT is trained only to predict final answers while RLVR generates full reasoning chains. This creates an asymmetric comparison that conflates two factors: (1) reasoning vs. direct prediction and (2) RL vs. supervised learning. A fair strategy is to include an SFT baseline trained on optimal reasoning chains (generated by the solver or sampled from successful RLVR rollouts). This would isolate whether gains come from RL exploration or simply having reasoning chains. | Q1: Can you provide results for SFT trained on reasoning chains (even if just for 7B/one level)?
>
> Thank you for raising this point. We agree it is useful to separate the effects of (1) fine-tuning on reasoning chains and (2) RL exploration. We therefore add ablation studies with 7B LLMs across all three levels by adding an SFT baseline trained on correct reasoning chains collected via rejection sampling (Sec. D.5).
>
> Setup. For each of the 8,000 training problems per level (association, intervention, counterfactual), we sample 32 chains-of-thought from Qwen2.5-7B-Instruct at temperature 1.0. We retain samples with correct final answers, and fine-tune on this rejection-sampled dataset (“rs32 sft”) for 5 epochs, selecting the best checkpoint by dev accuracy.
>
> Results. In both within-level and across-level generalization, rs32 sft outperforms cot init and direct-prediction sft, confirming that supervision on correct chains helps. However, rs32 sft still underperforms RLVR, especially on harder splits (full results in Tables 8 and 9).
>
> Error Analysis. We compared the reduction in probability errors, copy errors, and arithmetic errors in LLMs reasoning traces after SFT with rejection sampled reasoning chains or RLVR in Sec. D.8, and found that RLVR reduced these errors more so than SFT.
>
> Conclusion. This ablation suggests that RLVR’s gains come from both reasoning-chain supervision and its on-policy RL updates, rather than from fine-tuning on reasoning chains alone.
>
> Within-level Results:
> | Model | interv n10v2 easy (n=1086) | interv n10v2 medium (n=664) | interv n10v2 hard (n=348) | interv n10v2 all (n=2098) | assoc n10v2 easy (n=1684) | assoc n10v2 medium (n=1868) | assoc n10v2 hard (n=388) | assoc n10v2 all (n=3940) | counte n10v2 easy (n=24) | counte n10v2 medium (n=457) | counte n10v2 hard (n=231) | counte n10v2 all (n=712) |
> | --- | --- | --- | --- | --- | --- | --- | --- | --- | --- | --- | --- | --- |
> | 7b rl | **99.908** | **91.566** | **58.621** | **90.419** | **81.354** | **40.685** | **33.505** | **57.360** | **4.167** | **13.348** | **15.152** | **13.624** |
> | 7b rs32 sft | 96.593 | 70.331 | 21.552 | 75.834 | 69.418 | 19.647 | 14.433 | 40.406 | **0.000** | **10.941** | **9.957** | 10.253 |
> | 7b cot init | 61.510 | 51.958 | 16.092 | 50.953 | 29.513 | 11.135 | 6.701 | 18.553 | **4.167** | 8.753 | 4.762 | 7.303 |
> | 7b dp sft | 99.079 | 13.102 | 16.379 | 58.151 | 67.221 | 21.306 | 15.979 | 40.406 | **20.833** | **14.880** | **15.584** | **15.309** |
> | 7b dp init | 24.862 | 4.217 | 0.287 | 14.252 | 7.898 | 2.677 | 1.546 | 4.797 | **4.167** | 1.751 | 0.433 | 1.404 |

---

> ### Author Response · Authors · 2025-11-24
> **Response to W1 + Q1 Continued**
>
> Across-level Results by Test Level: (Training level indicated by system name, evaluation level indicated by column name)
> | Model | assoc n10v2 easy (n=1684) | assoc n10v2 medium (n=1868) | assoc n10v2 hard (n=388) | assoc n10v2 all (n=3940) |
> | --- | --- | --- | --- | --- |
> | rl 7b inte | 48.337 | **34.904** | **25.258** | 39.695 |
> | sft 7b rs32 inte | **67.043** | 24.732 | 17.784 | **42.132** |
> | rl 7b coun | 55.641 | 31.156 | **23.454** | **40.863** |
> | sft 7b rs32 coun | 43.943 | 22.645 | 15.464 | 31.041 |
> | 7b rl init | 29.513 | 11.135 | 6.701 | 18.553 |
> | sft 7b inte | 56.116 | 18.094 | **21.907** | 34.721 |
> | sft 7b coun | 31.116 | 18.737 | 18.299 | 23.985 |
> | 7b sft init | 7.898 | 2.677 | 1.546 | 4.797 |
>
> | Model | interv n10v2 easy (n=1086) | interv n10v2 medium (n=664) | interv n10v2 hard (n=348) | interv n10v2 all (n=2098) |
> | --- | --- | --- | --- | --- |
> | rl 7b asso | **98.619** | **87.500** | **52.874** | **87.512** |
> | sft 7b rs32 asso | 89.963 | 63.404 | 21.264 | 70.162 |
> | rl 7b coun | **99.079** | **87.199** | 44.540 | **86.273** |
> | sft 7b rs32 coun | 94.383 | 73.343 | 22.701 | 75.834 |
> | 7b rl init | 61.510 | 51.958 | 16.092 | 50.953 |
> | sft 7b asso | **99.263** | 19.729 | 18.391 | 60.677 |
> | sft 7b coun | 97.514 | 18.825 | 22.126 | 60.105 |
> | 7b sft init | 24.862 | 4.217 | 0.287 | 14.252 |
>
> | Model | counte n10v2 easy (n=24) | counte n10v2 medium (n=457) | counte n10v2 hard (n=231) | counte n10v2 all (n=712) |
> | --- | --- | --- | --- | --- |
> | rl 7b inte | **0.000** | **13.567** | **11.688** | 12.500 |
> | sft 7b rs32 inte | **4.167** | 11.597 | **10.390** | 10.955 |
> | rl 7b asso | **4.167** | **14.880** | **15.584** | **14.747** |
> | sft 7b rs32 asso | **4.167** | 10.503 | 7.359 | 9.270 |
> | 7b rl init | **4.167** | 8.753 | 4.762 | 7.303 |
> | sft 7b inte | **8.333** | 6.127 | 6.494 | 6.320 |
> | sft 7b asso | **8.333** | 9.409 | **9.957** | 9.551 |
> | 7b sft init | **4.167** | 1.751 | 0.433 | 1.404 |

---

> ### Author Response · Authors · 2025-11-24
> **Response to W2**
>
> > **(W2)** W2: The paper observes 3B models fail to benefit from RLVR and regress to direct prediction after training, but doesn't investigate what specifically these models lack.
>
> Thank you for asking us to clarify what the 3B models lack. We did analyze the reason that 3B models fail to benefit from RLVR and regress to direct prediction, but this may not have been presented clearly. We have now rewritten Analysis III (Sec. 4.3) to make this explicit. The key reason is that 3B’s initial reasoning chains are error prone - especially those that attempt any marginalization (Fig. 5 top). Here’s the relevant excerpt from Analysis III:
>
> > In fig. 5 (top), we also see that for 3b models, instead of learning to marginalize correctly, they learned to avoid marginalization after RLVR (grey), frequently predicting the answer without any calculations (see fig. 17 for an example). The high rate of failure (indicated by shading) of marginalization-based solutions (incremental, brute-force, and neighbors) prior to training is a possible explanation for why 3b LLMs avoided it after RLVR.
>
> Concretely, they frequently make derivation mistakes, e.g. mis-treat interventions as observations, apply incorrect marginalization formulas, and make basic arithmetic or copy mistakes. As a result, RLVR would assign  low reward to trajectories that attempt step-by-step reasoning, and the 3B model is pushed toward direct prediction instead. We added an example initial reasoning chain from the 3B model annotated errors in Fig.18 (which we reproduce below), illustrating its probability derivation errors and arithmetic errors.
>
> **Example: Causal Graph**
> > [...]
> > Here are the causal conditional probability tables (CPT) associated with the DAG:
> >
> > CPTs for v4:
> > P(v4) = [0.51, 0.49]
> >
> > CPTs for v8:
> > P(v8 | v4=0) = [0.02, 0.98]
> > P(v8 | v4=1) = [0.36, 0.64]
> >
> > CPTs for v7:
> > P(v7 | v8=0,v4=0) = [0.94, 0.06]
> > P(v7 | v8=0,v4=1) = [0.25, 0.75]
> > P(v7 | v8=1,v4=0) = [0.49, 0.51]
> > P(v7 | v8=1,v4=1) = [0.58, 0.42]
> >
> > CPTs for v2:
> > P(v2 | v8=0) = [0.11, 0.89]
> > P(v2 | v8=1) = [0.97, 0.03]
> >
> > CPTs for v9:
> > P(v9 | v4=0) = [0.95, 0.05]
> > P(v9 | v4=1) = [0.42, 0.58]
> >
> > CPTs for v3:
> > P(v3 | v9=0,v4=0) = [0.46, 0.54]
> > P(v3 | v9=0,v4=1) = [0.61, 0.39]
> > P(v3 | v9=1,v4=0) = [0.7, 0.3]
> > P(v3 | v9=1,v4=1) = [0.77, 0.23]
> >
> > CPTs for v6:
> > P(v6 | v7=0,v8=0) = [0.1, 0.9]
> > P(v6 | v7=0,v8=1) = [0.5, 0.5]
> > P(v6 | v7=1,v8=0) = [0.47, 0.53]
> > P(v6 | v7=1,v8=1) = [0.53, 0.47]
> >
> > CPTs for v5:
> > P(v5 | v3=0) = [0.38, 0.62]
> > P(v5 | v3=1) = [0.66, 0.34]
> >
> > CPTs for v0:
> > P(v0 | v3=0) = [0.96, 0.04]
> > P(v0 | v3=1) = [0.86, 0.14]
> >
> > CPTs for v1:
> > P(v1 | v4=0) = [0.59, 0.41]
> > P(v1 | v4=1) = [0.7, 0.3]
> >
> > Here's your Question: What is the marginal distribution of v2 given we intervented to set v8 to 1?
>
>
> **Example: Reasoning Chain of 3B Init**
>
> > THOUGHT PROCESS
> > To find the marginal distribution of v2 given that v8 is set to 1, we need to compute the probability distribution of v2 over its possible values (0 and 1) under the intervention. This involves marginalizing out the effect of v8 from the joint distribution of v2 and v8.
> >
> > The formula for the marginal distribution of v2 given v8=1 is:
> > \[ P(v2 = 0 | v8 = 1) = \sum_{v4} P(v2 = 0, v8 = 1 | v4) P(v4) \]
> > \[ P(v2 = 1 | v8 = 1) = \sum_{v4} P(v2 = 1, v8 = 1 | v4) P(v4) \]
> >
> > We need to compute the conditional probabilities \( P(v2 = 0, v8 = 1 | v4) \) and \( P(v2 = 1, v8 = 1 | v4) \) using the CPTs for v2 and v8.
> >
> > Let's start with \( P(v2 = 0, v8 = 1 | v4) \): **treated intervention as observation**
> > \[ P(v2 = 0, v8 = 1 | v4) = P(v2 = 0 | v8 = 1, v4) P(v8 = 1 | v4) \]
> > \[ P(v2 = 1, v8 = 1 | v4) = P(v2 = 1 | v8 = 1, v4) P(v8 = 1 | v4) \]
> >
> > From the CPTs:
> > \[ P(v2 = 0 | v8 = 1, v4) = 0.97 \]
> > \[ P(v2 = 1 | v8 = 1, v4) = 0.03 \]
> > \[ P(v8 = 1 | v4) = 0.64 \]
> >
> > Thus: **assumed v4 = 1 here and didn’t do v4 = 0**
> > \[ P(v2 = 0, v8 = 1 | v4) = 0.97 \times 0.64 = 0.6208 \]
> > \[ P(v2 = 1, v8 = 1 | v4) = 0.03 \times 0.64 = 0.0192 \]
> >
> > Now, we need to marginalize these over v4:
> > \[ P(v2 = 0 | v8 = 1) = \sum_{v4} P(v2 = 0, v8 = 1 | v4) P(v4) \]
> > \[ P(v2 = 1 | v8 = 1) = \sum_{v4} P(v2 = 1, v8 = 1 | v4) P(v4) \]
> >
> > Given the CPT for v4:
> > \[ P(v4 = 0) = 0.51 \]
> > \[ P(v4 = 1) = 0.49 \]
> >
> > So: **invalid marginalization formula and arithmetic error with 0.6208 × 0.49**
> > \[ P(v2 = 0 | v8 = 1) = 0.6208 \times 0.51 + 0.0192 \times 0.49 = 0.316608 + 0.009408 = 0.326016 \]
> > \[ P(v2 = 1 | v8 = 1) = 0.0192 \times 0.51 + 0.6208 \times 0.49 = 0.009792 + 0.299192 = 0.308984 \]
> >
> > Rounding to the nearest two digits:
> > \[ P(v2 = 0 | v8 = 1) \approx 0.33 \]
> > \[ P(v2 = 1 | v8 = 1) \approx 0.31 \]
> >
> > ANSWER
> > [0.33, 0.31]<|im_end|>
> >

---

> ### Author Response · Authors · 2025-11-24
> **Response to W3, W4, and W5**
>
> > **(W3)** W3.1: The reward is simply r = 0.8 · accuracy + 0.2 · format with threshold t=0.01. Is the 0.8/0.2 weighting optimal?
>
> Thank you for raising this question. Reward functions that combine a format reward and a correctness reward have been used in prior work (e.g. [1, 2]). We did not optimize this reward ratio and used (0.2, 0.8) throughout all of our experiments, we chose the particular values to encode our prior of emphasizing correctness. To check the effect of this particular hyper-parameter choice, we added two runs with 7B RLVR on the association level with reward weighting (1.0, 0.0) and (0.6, 0.4), and found no significant difference in downstream performance compared to the (0.8, 0.2) weighting used in our main experiments. The results are included below (systems not significantly different worse than the best is bolded).
>
> | Model | assoc n10v2 easy (n=1684) | assoc n10v2 medium (n=1868) | assoc n10v2 hard (n=388) | assoc n10v2 all (n=3940) |
> | --- | --- | --- | --- | --- |
> | rl 7b asso (0.8, 0.2) | 81.354 | **40.685** | **33.505** | **57.360** |
> | rl 7b asso (1.0, 0.0) | **83.017** | **41.435** | **30.928** | **58.173** |
> | rl 7b asso (0.6, 0.4) | **82.779** | **40.525** | **35.825** | **58.122** |
>
> > **(W3)** W3.2: Would shaped rewards (partial credit for intermediate steps) help?
>
> Thank you for raising this important question. Partial credit for intermediate steps could for example be learned value functions (e.g. in PPO) or intermediate correctness functions (e.g. process reward models). They are helpful in some situations [3] but can be tricky to adopt effectively in practice, due to the expensive costs associated with training and deploying these intermediate value/reward models [5 or to collect data for them [4], and also due to reward hacking with learned intermediate value/reward models [1]. We also note that our focus in this paper is studying the generalization of RLVR for fine-tuning reasoning LLMs, which in practice is often based on outcome rewards without process supervision [1, 2].
>
> > **(W4)** W4: The writing is vague. For example, the description of the third finding "What did RLVR learn? " is too abstract, containing too many technical terms. They should present this result in a concrete style.
>
> Thank you for helping us improve the clarity of our writing. We have rewritten our third finding to discuss more concrete details and refer to examples. We refer the reviewer to our revised draft due to character limits of the rebuttal form.
>
> > **(W5)** W5: Why not compare RLVR with other RL-based post-training paradigm, such as RLHF and RLAIF?
>
> Thank you for raising this question. We’d like to clarify again that our main focus in this paper is to study the generalization behavior of RLVR itself when used to train reasoning LLMs. On this topic, prior work has often compared RLVR with SFT baselines [6,7] - when a verifiable reward function is available, it can be much cheaper than RLHF which depends on human annotations that are costly and time-consuming to collect, and it can be less hackable than RLAIF.
>
> [1] D. Guo et al., ‘DeepSeek-R1 incentivizes reasoning in LLMs through reinforcement learning’, Nature, vol. 645, no. 8081, pp. 633–638, Sept. 2025.
>
> [2] Biomni and T. Sky RL, ‘Biomni-R0: Using RL to Hill-Climb Biomedical Reasoning Agents to Expert-Level’, Biomni - A General-Purpose Biomedical AI Agent. Sept-2025.
>
> [3] J. Cheng et al., ‘Stop Summation: Min-Form Credit Assignment Is All Process Reward Model Needs for Reasoning’, in The Thirty-ninth Annual Conference on Neural Information Processing Systems, 2025.
>
> [4] H. Lightman et al., ‘Let’s Verify Step by Step’, in The Twelfth International Conference on Learning Representations, 2024.
>
> [5] Z. Shao et al., ‘DeepSeekMath: Pushing the Limits of Mathematical Reasoning in Open Language Models’, arXiv [cs.CL]. 2024.
>
> [6] T. Chu et al., ‘SFT Memorizes, RL Generalizes: A Comparative Study of Foundation Model Post-training’, in Forty-second International Conference on Machine Learning, 2025.
>
> [7] H. Chen et al., ‘SFT or RL? An Early Investigation into Training R1-Like Reasoning Large Vision-Language Models’, arXiv [cs.CL]. 2025.

---

> ### Author Response · Authors · 2025-11-24
> **Response to Q2**
>
> > **(Q2)** Can models solve trivially simple counterfactuals (e.g., 3-node graphs, no marginalization needed)?
> Thank you for raising this question. We agree it would be interesting to look at simpler counterfactual problems to understand the source of their challenge. We added an evaluation of all our trained models on a simpler counterfactual dataset - CLadder dataset’s deterministic counterfactual subsplit. These problems are based on smaller causal graphs (three to four nodes) and have deterministic mechanisms that make marginalization unnecessary. We converted a subset of CLadder det-counterfactual split into our data format, and evaluated all of our counterfactually fine-tuned models. We added a detailed discussion in Sec. D.6 which we summarize below
>
> Setup. Evaluate SFT and RL models on 790 questions from the CLadder det-counterfactual subsplit [8], converted into our data format.
>
> Results. Compared to the simple split of our counterfactual test set, our models perform much better on the simpler det-counterfactual questions.
>
> Conclusion. Models can solve trivially simple counterfactuals reasonably well, indicating marginalization over the posterior of exogenous variables contributes to the challenge of our counterfactual level task.
> | Model | cladder det-counterfactual (n=790) |
> | --- | --- |
> | rl 32b | **99.747** |
> | rl 32b curriculum | **99.747** |
> | rl 32b init | 98.987 |
> | rl 7b | 84.937 |
> | rl 7b curriculum | 84.937 |
> | rl 7b init | 67.468 |
> | rl 3b | 64.051 |
> | rl 3b curriculum | 57.722 |
> | rl 3b init | 40.506 |
> | sft 32b | 70.633 |
> | sft 32b init | 81.392 |
> | sft 7b | 61.139 |
> | sft 7b init | 43.038 |
> | sft 3b | 48.734 |
> | sft 3b init | 47.089 |
>
> | Model | counte n10v2 (n=24) |
> | --- | --- |
> | rl 32b | 4.167 |
> | rl 32b curriculum | 0.000 |
> | rl 32b init | 0.000 |
> | rl 7b | 4.167 |
> | rl 7b curriculum | 0.000 |
> | rl 7b init | 4.167 |
> | rl 3b | 4.167 |
> | rl 3b curriculum | 0.000 |
> | rl 3b init | 0.000 |
> | sft 32b | **29.167** |
> | sft 32b init | 4.167 |
> | sft 7b | **20.833** |
> | sft 7b init | 4.167 |
> | sft 3b | 0.000 |
> | sft 3b init | 0.000 |
>
> [8] Z. Jin et al., ‘CLadder: A Benchmark to Assess Causal Reasoning Capabilities of Language Models’, in Thirty-seventh Conference on Neural Information Processing Systems, 2023.

---

> > ### Comment · Reviewer_yW6b · 2025-11-28
> >
> > Thank the authors for the comprehensive response. My concerns have been well addressed. I am willing to increase my rating to 6.

---

### Comment · Area_Chair_Af9R · 2025-11-27

Dear Authors and Reviewers,

The discussion phase will end soon. If you want to further discuss comments and replies with each other, please post your thoughts by adding official comments.

Thanks for your efforts and contributions to ICLR 2026.

Best regards,

Your Area Chair

---

### Author Response · Authors · 2025-12-03
**Response Summary**

We thank the reviewers again for their informative feedback and thank the AC for managing this unexpected situation. Below we briefly summarize our responses to each reviewer’s concerns:

**Reviewer  yW6b** (score 4 → 6)
| Concern      | Response      |
|--------------|---------------|
| “SFT is trained only to predict final answers while RLVR generates full reasoning chains. This creates an asymmetric comparison that conflates two factors: (1) reasoning vs. direct prediction and (2) RL vs. supervised learning.” |  We added results for SFT on reasoning chains, at all levels, for the 7B model size. Results show that RLVR > SFT on reasoning > SFT on direct prediction, showing that RL is helpful on top of reasoning. |
| “The paper … doesn't investigate what specifically [3B] models lack.” | We clarified that we did indeed investigate what 3B models lack that led to the ineffectiveness of RLVR in analysis-III, and we improved its presentation. We found that prior to fine-tuning, 3B models make many derivation errors during step-by-step marginalization, which likely led to consistent negative reinforcement whenever it was attempted, pushing the model towards direct prediction instead. |
| “Is the 0.8/0.2 weighting optimal? Would shaped rewards (partial credit for intermediate steps) help?”  | We added results for 7B models on the association level with (0.0, 1.0) and (0.4, 0.6) reward weighting and found limited change in final performance. We also clarified that while intermediate rewards may help, they can be hard to adopt due to reward hacking and cost of running and training intermediate reward models. Also, RLVR, the focus of our study and the current paradigm for training reasoning LLMs, often uses outcome rewards in practice. |
| “The writting is vague. For example, the description of the third finding “What did RLVR learn?” is too abstract …”| We rewrote analysis-III to be more concrete and expanded examples. |
| “Why not compare RLVR with other RL-based post-training paradigm, such as RLHF and RLAIF?”| We clarified that prior work [1,2] studying RLVR has often focused on SFT as baselines, and that when verifiable reward is available it is often cheaper to get data for than RLHF, and less hackable than RLAIF. |

[1] T. Chu et al., ‘SFT Memorizes, RL Generalizes: A Comparative Study of Foundation Model Post-training’, in Forty-second International Conference on Machine Learning, 2025.

[2] H. Chen et al., ‘SFT or RL? An Early Investigation into Training R1-Like Reasoning Large Vision-Language Models’, arXiv [cs.CL]. 2025.

---

> ### Author Response · Authors · 2025-12-03
> **Response Summary - Continued**
>
> **Reviewer  5AJc** (score 6 → 8)
> | Concern      | Response      |
> |--------------|---------------|
> | “The RLVR experiments use 7.5K and 2.5K samples, while the SFT model is trained on 5K samples. This discrepancy makes the quantitative comparison between models less reliable. I suggest adding an ablation study where models (or checkpoints) are trained on the same amount of data and for comparable GPU hours to mitigate this concern.” | We clarified that this was a misunderstanding, and the amount of data was the same across methods. Compute-wise, the SFT models use less compute than RL because it doesn’t need to generate rollouts. The SFT models are trained for 5 epochs, where dev loss already exhibits a U-shape curve, and we pick the best checkpoint by dev loss. As for RL models, we train 7B and 32B models with roughly matched compute, but we used a quarter of that compute on 3B models because their dev performance plateaued quickly. We added results for 3B models trained with quadrupled compute to match 7B and 32B models: while there is a slight increase in performance, the comparisons with other models and our findings about its behavior remain unchanged. |
> | “It would strengthen the paper to include a detailed error analysis comparing SFT and RLVR models on these sub-tasks [retrieving correct numerical values and performing basic computations].” | We added an analysis of copy errors and arithmetic errors. |
>
> **Reviewer Cf3X** (score 4 → 6)
> | Concern      | Response      |
> |--------------|---------------|
> | “The current version seems a bit colloquial rather than a formal academic paper.”| We rewrote our abstract to improve formality and clarity. |
> | “Similarly, …, I would encourage authors to revise the introduction throughly.” | We rewrote our introduction to improve formality and clarity. |
> | “I would suggest that authors include a discussion on the practical value of the studied formal causal reasoning. Since I believe LLMs are more targeted at the commonsense setting, and there already exist lots of formal causal tools in the area of causal inference.” | We clarified that our task is designed with studying RLVR generalization in mind, which benefits from the existence of formal tools — they enable us to systematically generate training and test data with solutions generated by formal solvers. We added that even though our data and task are more formal and abstract, it could be used to probe and train subskills useful for a LLM when solving more practical causal reasoning tasks. |
> | “Authors should consider assigning a formal name for their datasets.” | We considered their suggestion of RLCausal. |
> | “I would suggest that authors add an individual section on the differences between their newly proposed datasets and existing benchmarks. Why can't other benchmarks test the generalization abilities of RLVR?” | We clarified that our goal of probing RLVR generalization calls for focusing on the more formal, step-by-step parts of causal reasoning, which led us to pick probabilistic inference in causal graphical models as a task. We also wanted a large and diverse set of graphs and queries to probe RLVR’s generalization across different query difficulties. These specific requirements were not simultaneously available in existing datasets so we created our own. |
> | “Authors should revise Figure 1 to a more abstract and summarized representation.” | We improved the clarity of the example in figure 1 and improved colors. |
>
> We once again thank the reviewers and AC for their time and feedback. We believe the added experiments, analyses, and clarifications sufficiently addressed the main concerns raised by the reviewers and helped strengthened our paper, and that our work advances the understanding of the emerging paradigm of RLVR and the causal reasoning ability of LLMs.

---

### Meta-Review · Area_Chair_GYqK · 2025-12-26

**Summary:**

This paper investigates when and why RLVR has a better generalization ability than SFT in LLM reasoning. It uses probabilistic reasoning in causal graph models to construct a controlled test platform for association, intervention, and counterfactual queries. It proposes a benchmark called RLCausal, which clearly distinguishes the skills of derivation and calculation from language models, and evaluates RLVR variants and SFT through additional trajectory analysis.

The advantages of this paper include the formulation of the research problem and experimental control. Causal reasoning solves problems of precise and scalable difficulty, and the evaluation is diagnostic rather than purely performance-driven. The extensive empirical results demonstrate the key findings: the benefits of RLVR rely on sufficient underlying model capabilities and tend to improve system marginalization behavior while reducing inference errors; however, counterfactual reasoning remains challenging, even in scaled situations.

As for the disadvantages, reviewers raised concerns about the comparison of baselines and compute, the explanation for the 3B collapse, the reward design, and the clarity of presentation. In response, the authors implemented a stronger SFT baseline (trained on correct reasoning trajectories) and showed that "RLVR > SFT on reasoning trajectories > direct-prediction SFT", indicating that the gains are not only due to exposing trajectories. The authors' responses clarified computational costs, conducted a higher-compute 3B run, and expanded the scope of RL's reward designs. In addition, the authors explained the collapse of the 3B model and improved the writing and contextualization.

Overall, after strengthening the baseline and clarifying the issues, the remaining concerns have been well addressed, and all three reviewers increased their scores. This work presents a rigorous and insightful study of RLVR generalization in a controlled but meaningful reasoning environment. I recommend Accept (Poster).

**Reviewer Concerns:**

Reviewers raised concerns about the comparison of baselines and compute, the explanation for the 3B collapse, the reward design, and the clarity of presentation. These concerns are well addressed by the authors’ responses.

**Reviewer Scores:**

Three reviewers submitted their reviews (with scores 4,4,6), and all increased their scores to acceptance (6,6,8).

---

### Decision · Program_Chairs · 2026-01-26

Accept (Poster)